biomedical engineering/biomathematics

aneurysm rupture risk prediction, geometry feature, machine learning

**Authors for correspondence:**
Shixin Xu
e-mail: shixin.xu@dukekunshan.edu.cn
Yibin Fang
e-mail: fangyibin@vip.163.com
Huaxiong Huang
e-mail: hhuang@uic.edu.cn

†These authors contributed equally to this study.

# Predicting the risk of rupture for vertebral aneurysm based on geometric features of blood vessels

Shixuan Li[1,†], Ruiqi Pan[2,†], Arvind Gupta[1], Shixin Xu[3], Yibin Fang[4] and Huaxiong Huang[5,6,7]

[1]Department of Computer Science, University of Toronto, Toronto, Ontario, Canada
[2]Department of Computer Science, McCormick School of Engineering, Northwestern University, Evanston, IL, USA
[3]Data Science Research Center, Zu Chongzhi Center for Mathematics and Computational Sciences, Duke Kunshan University, Kunshan, Jiangsu, People's Republic of China
[4]Department of Neurosurgery, Changhai Hospital, Shanghai, China
[5]Research Centre for Mathematics, Advanced Institute of Natural Sciences, Beijing Normal University, Zhuhai, People's Republic of China
[6]BNU-HKBU United International College, Zhuhai, People's Republic of China
[7]Department of Mathematics and Statistics, York University, Toronto, Ontario, Canada

SX, 0000-0002-8207-7313

A significant proportion of the adult population worldwide suffers from cerebral aneurysms. If left untreated, aneurysms may rupture and lead to fatal massive internal bleeding. On the other hand, treatment of aneurysms also involve significant risks. It is desirable, therefore, to have an objective tool that can be used to predict the risk of rupture and assist in surgical decision for operating on the aneurysms. Currently, such decisions are made mostly based on medical expertise of the healthcare team. In this paper, we investigate the possibility of using machine learning algorithms to predict rupture risk of vertebral artery fusiform aneurysms based on geometric features of the blood vessels surrounding but excluding the aneurysm. For each of the aneurysm images (12 ruptured and 25 unruptured), the vessel is segmented into distal and proximal parts by cross-sectional area and 382 non-aneurysm-related geometric features extracted. The decision tree model using two of the features (standard deviation of eccentricity of proximal vessel, and diameter at the distal endpoint) achieved 83.8% classification accuracy. Additionally, with support vector machine and logistic regression, we also achieved 83.8% accuracy with another set of two features (ratio of mean curvature between distal and proximal parts, and diameter at the distal endpoint). Combining the aforementioned three features with

integration of curvature of proximal vessel and also ratio of mean cross-sectional area between distal and proximal parts, these models achieve an impressive 94.6% accuracy. These results strongly suggest the usefulness of geometric features in predicting the risk of rupture.

# 1. Introduction

Each year, almost 500 000 deaths are caused by cerebral aneurysms worldwide [1]. Aneurysms, which occur upon pathological dilation of blood vessel and weakening of the vessel wall, may grow and rupture without proper treatment [2–4]. While the resulting haemorrhage may be lethal, treatment is also associated with risk and undesired consequences that are not negligible [5]. Therefore, surgeons often face a dilemma due to the significant risks involved. To assist surgical decision-making, it is highly desirable to identify rupture risks that are complementary to expert knowledge of the healthcare professions.

Previous studies on haemodynamics of aneurysms have found that the parent vessel of the aneurysm is strongly associated with haemodynamics characteristics, namely, curved parent vessels result in adverse haemodynamics [6,7]. This in turn contributes to unstable and complex flow patterns within and near the aneurysm, increasing the risk of rupture [8]. Specifically, based on computational fluid dynamics (CFD) analysis, low wall shear stress (WSS) along with parallel WSS vectors and lower flow velocity are associated with thin aneurysm wall and thus a higher risk of rupture [9]. Correspondingly, by combining machine learning with CFD simulations, a recent study [10] identified aneurysm location, mean surface curve and maximum flow velocity as important predictors for the risk of rupture of cerebral aneurysms. Our study, therefore, focuses on how geometric features of vessels impact the risk of rupture from a data-driven approach and potentially serves as a complement to the haemodynamics models.

Based on their shape, aneurysms are categorized into saccular and fusiform [5]. Previous studies have shown the importance of haemodynamic and morphological evaluation in predicting the rupture of saccular aneurysms [11–16]. As for fusiform aneurysms, morphological study may also play an important role in predicting the risk of rupture [17,18]. However, with the higher complexity of their morphological characteristics, fusiform aneurysms, specifically vertebral artery fusiform aneurysms (VAFA), require more investigation and thus are the focus of this study.

Zhao *et al.* [5] proposed a model based on five geometric features of VAFA morphology and achieved state-of-the-art classification performance ($81.43 \pm 13.08\%$). The feature generation procedure and selected features depend heavily on the information of the aneurysm itself. Specifically, the size of the aneurysm is used to do the segmentation. However, the shape of an aneurysm often changed when it ruptured, and intra-aneurysmal thrombus can also conceal the overall perspective, which both made the results of CFD analysis based on retrospective collected data less valuable. In spite of the change of aneurysm shape, the rupture rarely changes the parent artery and the relationship between the parent and daughter arteries, which could actually be the cause of the growth and the rupture of the aneurysm. At the same time, in the top 5 identified features, only the third one is the aneurysm-specific feature, solidity of aneurysm, and the other four are the features related to the parent vessel. The objective of this paper is to predict the risk of rupture of VAFA based on the geometric properties of blood vessels. Using the cross-sectional area change of the vessel, we proposed a new parent artery segmentation method to generate features from data points. Then a variety of machine learning techniques for feature selection and models are proposed based on different sets of geometric features.

The rest of the paper is organized as follows. In §2, we present detailed information about the used dataset as well as the methods for segmentation, feature extraction, feature selection, modelling and evaluation. Selected feature sets of varying sizes, along with the evaluation results of fitting each to four machine learning models, are proposed in §3. Section 4 is contributed to provide a detailed discussion of the selected features.

# 2. Methods

## 2.1. Patients and image data

A total of 37 patients (23 males, 14 females, mean age = 52.43 years, s.d. = 10.12) with 12 ruptured and 25 unruptured aneurysms were included in the study. All three-dimensional images in stereolithography (STL) format of vertebral artery aneurysm images are processed and geometric features are extracted,

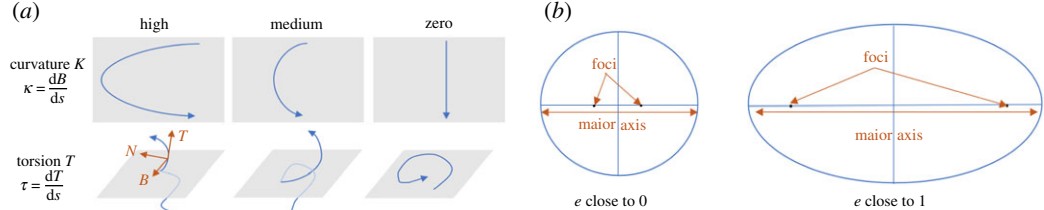

**Figure 1.** Panel (*a*) illustrates curvature and torsion, and (*b*) illustrates eccentricity.

following the procedure in [5]. The images are initially processed by MeshLab [19], an open source software, to derive geometric characteristics. The complete definitions of the geometric characteristics are listed in electronic supplementary material, appendix A.

## 2.2. Defining geometric characteristics

An important characteristic for modelling the blood vessel, the centreline of the vessel is defined using the same method as the previous research, which queues discrete points and connects the data points by three-dimensional curve fitting algorithm [20,21].

Tangent unit vectors, normal unit vectors and binormal unit vectors are then calculated based on the centreline. The curvature and torsion are evaluated by using Frenet–Serret formulae. Additional geometric indices are then generated, including maximum, minimum and equivalent diameters (the diameter of a circle with the same area as the region), cross-sectional area, area change rate, eccentricity, solidity (the proportion of the pixels in the convex hull) and extent (the ratio of pixels in the region to pixels in the total bounding box) [5]. Definitions of several important geometric indices are listed below and illustrated in figure 1. More detailed explanation of the geometric indices can be found in electronic supplementary material, appendix A.

1. Centreline curvature $\kappa$ (mm$^{-1}$):
   Curvature $\kappa$ represents the rate of change of the angle through which the tangent to a curve turns in moving along the curve. The curvature $\kappa$ is the solution of the Frenet–Serret formulae. See electronic supplementary material, appendix, figure A.1 for an example of curvature.
2. Centreline torsion $\tau$ (mm$^{-1}$):
   In three dimensions, the torsion $\tau$ of a curve measures how sharply it is twisting out of the plane of curvature. The torsion $\tau$ is solved by the Frenet–Serret formulae. See electronic supplementary material, appendix, figure A.1 for an example of torsion.
3. Cross-sectional Area $A$ (mm$^2$):
   Cross-sectional area of the blood vessel, calculated by the number of pixels.
4. Eccentricity $e$ (unitless):
   The ratio of the distance between the foci of the ellipse and its major axis length MaxD (electronic supplementary material, appendix, figure A.3). Eccentricity $e =$ distance between foci/length of the major axis. $e$ is close to 0 if the shape is closer to a circle, close to 1 if more elliptical.

## 2.3. Segmentation

The haemodynamic [8] plays an essential role in the rupture of brain aneurysm. The geometric properties of the proximal vessels impact the normal blood flow, and those of the distal vessels also reflect such change upstream [22]. In order to understand how these parts of the vessel relate to the risk of rupture, the vessel is split into five different segments (two proximal, two distal). The motivation behind such segmentation is that the location where the change in blood vessel geometric properties occurs could potentially relate to how it affects the aneurysm. However, the criterion for previous segmentation, aneurysm length, uses aneurysm-related data. As per our goal of excluding aneurysm-related information from the entire segmentation and feature selection process, new criteria are created and the blood vessel is re-segmented in a patient-specific way.

The segmentation process uses points on the centreline curve. We set each segment to have an equal number of curve-fitting points. The thresholds that determine the length of segmentation are set by the standard deviation of cross-sectional area. Standard deviation is used here as a statistically meaningful point that determines whether the difference between points and the mean is noteworthy. Our

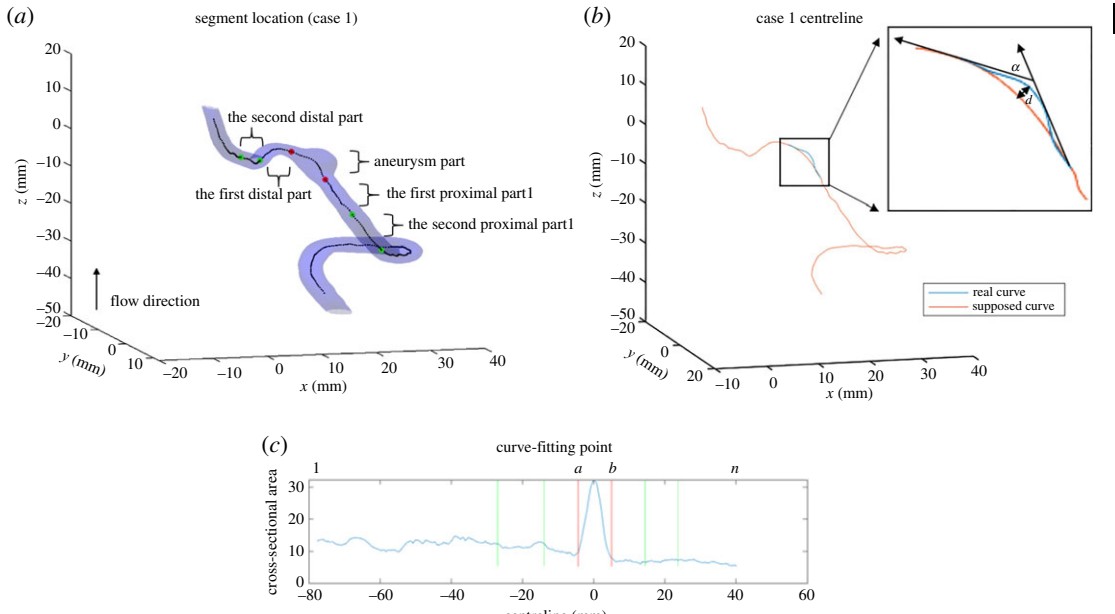

**Figure 2.** The subfigure (*a*) demonstrates segment location of five segments, including the first/second proximal part, the first/second distal part and the aneurysm part, all with same number of curve-fitting points. (*b*) Demonstrates the angle (*α*) and distance (*d*) between two centreline tangent vectors at the neck of the bulge. The subfigure (*c*) illustrates the segmentation algorithm, specifically where the points of segmentation locate relative to cross-sectional area along the curve.

approach is similar to the method for detecting change points for time-series data. The detailed algorithm is summarized as follows (also see figure 2*c*). Define:

— $a$ = label of aneurysm starting point;
— $b$ = label of aneurysm end point;
— $s$ = the number of curve fitting points within a segment;
— $n$ = the total number of curve fitting points for this individual;
— $ps$ = cross-sectional area at start of aneurysm;
— $pe$ = cross-sectional area at the end of aneurysm.

To calculate $s$, we further define:

— baseline = $ps$ (if the point of interest is in the proximal vessel) and $pe$ (if the point of interest is in the distal vessel);
— threshold = standard deviation of cross-sectional area of the entire vessel (excluding the aneurysm part);
— default$_s$ = min $((a-1)/2, (n-b)/2)^1$.

## 2.4. Feature generation

Based on the segmentation, we generated three groups of features: (i) the maximum, mean, standard deviation, integration and variation of the geometric index in each segment; (ii) the ratios of features in (i) between different segments; and (iii) the angle (*α*) and distance (*d*) between two centreline tangent vectors at the neck of the bulge (figure 2*b*). We generated a new dataset ($37 \times 384$) with less features than those in [5] ($37 \times 563$). A complete list of features can be found in electronic supplementary material, appendix A.

## 2.5. Feature selection

Excluding the outcome variable and the identifier variable, the generated dataset has 382 geometric features of the blood vessels. As the first step, an initial screening of the features was done by doing individual *T*-tests; the 31 significant features with a *p*-value less than or equal to 0.1 were selected to avoid overfitting due to the limit number of observations.

---

[1]the maximum number of points a segment can have.

---

**Algorithm 1:** Calculation of s

---

1: **INPUT**: Centerline datapoints: $\mathcal{D}$, and all parameters above;
2: scan the proximal vessel to find the first point whose difference with baseline cross sectional area exceeds the threshold : $p1$ ;
 **for** *point i in D[a:-1:1]* **do**
 | find the first point p1 in the proximal vessel that satisfies |p1-baseline| $\geq$ threshold ;
 **end**
3: scan the distal vessel to find the first point whose difference with baseline cross sectional area exceeds the threshold : $d1$ ;
 **for** *point i in D[b:1:n]* **do**
 | find the first point d1 in the distal vessel that satisfies |d1-baseline| $\geq$ threshold ;
 **end**
4: calculate s based on p1 and d1 ;
 **if** *p1 and d1 both can be found* **then**
 | s = min(a-p1,d1-b) ;
 **end**
 **if** *only p1 can be found and (a-p1)* $\leq$ *default$_s$* **then**
 | s=a-p1 ;
 **end**
 **if** *only d1 can be found and (d1-b)* $\leq$ *default$_s$* **then**
 | s=d1-b ;
 **end**
 **if** *neither p1 nor p2 can be found* **then**
 | s=default$_s$ ;
 **end**
5: **OUTPUT**:
 the number of points for each segment = s, the label of the first proximal point = a-s,
 the label of the second proximal point = a-2s,
 the label of the first distal point = b+s,
 the label of the second distal point = b+2s ;

---

The selected subset was passed into the following four different feature selection methods:

1. selection of features with largest *t*-statistic values (i.e. smallest *p*-values);
2. recursive feature elimination with support vector machine (SVM) being the estimator;
3. recursive feature elimination with logistic regression (LR) being the estimator;
4. recursive feature elimination with random forest (RF) being the estimator.

In order to partially address the concern of overfitting, we repeated the feature selection process 30 times, each time on a randomly selected subset of 80% of the data. We are interested in the features that are selected frequently enough (e.g. at least two-thirds of the times), since this indicates more consistency and less dependency on the training set. Owing to the small sample size, the feature set size should not be larger than five. We constructed feature sets of varying sizes to explore the possibility of having a smaller feature set while maintaining a reasonable amount of predictive power.

We used the `scikit-learn` Python package for implementation, specifically `SelectKBest` and `RFE` from `sklearn.feature_selection`. We specified the estimator as described above and the number of features to select as 5, and we accepted the default value of all the other parameters.

It should be noted that a common option is to combine feature selection and model training when using RFE. However, in this specific case, directly inputting the feature set to the same model as the one used by RFE could lead to over-optimistic results. Hence, we decided to separate feature selection and model training into two stages.

### 2.5.1. Recursive feature elimination

Recursive feature elimination (RFE) is a wrapper feature selection method that requires specification of an estimating model. It uses a backward feature selection process: the algorithm starts with fitting the model with all features and eliminates the feature(s) with the least importance; then it fits the model again and repeats the process until a feature set of desired size is generated.

The importance of the features is determined differently depending on the specified model. For instance, for support vector machine and logistic regression, it uses the estimated coefficients; whereas for decision tree and random forest, it uses the estimated feature importance values.

## 2.6. Models

### 2.6.1. Support vector machine

Support vector machine (SVM) is a commonly used supervised model for classification. For this study, we used SVM with a linear kernel, which assumes that there exists a hyperplane that separates the outcome classes.

The Python library function `sklearn.svm.SVC` was used, with all parameters other than `kernel` kept at their default value.

### 2.6.2. Logistic regression

Logistic regression (LR) is another linear model widely used for binary classification problems. It assumes that there exists a linear relationship between the log odds of the outcome being positive (e.g. odds that an aneurysm ruptures) and the predictors. $L_2$ penalty is also used to control the size of the coefficients and to avoid overfitting.

Most of the default settings of `sklearn.linear_model.LogisticRegression` were respected; the only change made was to set `solver='liblinear'`, which is suitable for small datasets according to the `scikit-learn` documentation.

### 2.6.3. K-nearest neighbour

K-nearest neighbour (KNN) classifies a data point based on the labels of its $k$ nearest data points using majority vote. That is, given a $k$-value and a test data point, if at least $\lceil \frac{k}{2} \rceil$ of its $k$ nearest data points are positive, then the test data point is classified as positive; otherwise, it is classified as negative.

In the actual implementation, we chose the default settings of `sklearn.neighbors.KNeighbors Classifier`. Specifically, we used $k = 5$ and specified the commonly chosen Euclidean distance as the distance measure.

### 2.6.4. Decision tree

Decision tree is a flexible nonlinear classifier which does not assume a specific relationship between the outcome and the predictors and automatically takes interaction between predictors into consideration. It recursively splits the data points by an optimal threshold of a selected feature; the feature and the threshold used in each split are chosen so that a specified measure (e.g. Gini coefficient) is optimized. The same feature can be chosen for different splits.

It is always possible to keep splitting the data until all points are correctly classified. However, this leads to overfitting. One effective way to prevent this is to control the maximum tree depth. As mentioned, we limited our feature set size to 5. We aimed to minimize the possibility of overfitting while granting a sufficient amount of freedom to the selection process. A maximum tree depth of 1 would be too restrictive since it forces the algorithm to select at most one feature and do one split. Similarly, a maximum tree depth of 2 implies a total of at most three features/splits. Therefore, we set a limit on the tree depth such that using each feature once would be still feasible while overfitting could be avoided as much as possible. For feature sets of size 4 or 5, we set the maximum tree depth to be 3; for smaller feature sets, we set the maximum tree depth to be 2.

Hence, we used `sklearn.tree.DecisionTreeClassifier` with its `max_depth` adjusted to the corresponding value.

## 2.7. Evaluation

Each feature set is fitted to each of the four aforementioned models. To evaluate and compare the prediction performance, two methods were used. In the first approach, mean accuracy from leave-one-out (LOO) cross validation was computed. In the second method, we repeated 100 rounds of 80% train and 20% test split and computed the mean accuracy.

# 3. Results

We experimented with different numbers of segments for comparison. We segmented the vessel containing aneurysm into three segments (one distal and one proximal) based on curvature and

torsion, and into seven segments (three distal and three proximal) based on cross-sectional area (see electronic supplementary material, appendix, figure A.4). We found that five segments (two distal and two proximal) would achieve the best balance. We present the feature selection and evaluation result using five-segment dataset. The meaning of the five most important features is described below.

## 3.1. Feature selection

Table 1 presents the feature selection results. In general, the results are reasonably consistent across different selection methods.

Based on the summary in the table, we can make the following observations. Restricting to the top 2 features selected by each method would give us a feature set of size 5, consisting of the following.

1. Curvature_mean_PD: ratio of mean curvature between proximal 1 and distal 1;
2. Diameter_normalD2: the diameter at endpoint of distal 2 (mm);
3. Eccentricity_std_P2: the standard deviation of eccentricity in proximal 2;
4. Curvature_integration_P: integration of curvature in proximal 1;
5. CrossArea_mean_PD2: ratio of mean cross-sectional area between proximal 2 and distal 2.

Taking the features selected at least two-thirds of the times, would give us a feature set of size 4, with CrossArea_mean_PD2 excluded from the previous set. Choosing among the consistently selected features, we constructed smaller feature sets to avoid overfitting. Specifically, we considered one feature set of size 3 with Curvature_mean_PD, Diameter_normalD2 and Eccentricity_std_P2, as well as the three pair-wise combinations of these three features. Finally, it is worth noting that the Pearson correlation coefficient matrix in figure 3 confirmed that the correlation between the selected features is low.

## 3.2. Model validation

The prediction performance of the six feature sets using the four described models is presented in table 2. Depending on the feature set, the best-performing model is different.

The decision tree model performs the best on the set with four features; also, with the set with just Diameter_normalD2 and Eccentricity_std_P2, the leave-one-out (LOO) accuracy achieves 83.8%. Note that during the LOO evaluation process, many different decision trees are fitted on different subsets of the 37 cases and so the decision boundaries would vary.

For illustration purpose, here we show the decision tree fitted using all 37 cases. Since the depth is two, there are three non-leaf nodes and four leaf nodes. Here is how the results can be understood and interpreted.

1. In each non-leaf node, the first line is the criterion to do the classification. For instance, if the diameter of the vessel at the further endpoint of distal part 2 is less or equal to 2.947, then we move onto its left child node. Otherwise if that criterion is not satisfied, we move onto its right child node.
2. The 'samples' in a node is the total number of cases that reach the node; 'value' indicates (the number of unruptured cases, the number of ruptured cases); 'class' is decided by the majority rule using 'value'. The very first node has 37 cases to start with. Then samples = 13 in its left child means that 13 of the 37 cases satisfy the first criterion. Among the 13 cases, 3 are unruptured and 10 are ruptured, thus the class of this node is ruptured. It is useful to have a class label for non-leaf nodes as well: if we decide not to ask further questions and stop at a non-leaf node, then that class label will be our prediction.
3. The orange-ish colours indicate class=unruptured; the blue-ish colours indicate class=ruptured. The darker the colour, the more confident the prediction is (i.e. the ratio of the majority class to the minority class is larger).

As shown in figure 4a, three of the four leaf nodes are classified as unruptured; it indicates that a case is predicted to be ruptured if and only if its diameter is less than or equal to 2.947 and eccentricity std in proximal 2 is greater than 0.071. In other words, all cases inside the blue region as shown in figure 4b are classified as ruptured. The training accuracy is better than the cross-validation accuracy because here the tree is fitted to the entire set. Similarly, the decision tree below is fitted using the sets of four features and has depth three. It has the same interpretation as we described for the previous smaller tree.

**Table 1.** Frequency table of top 6 features selected by the four methods; top 2 are highlighted in italics and frequent features (selected at least two-thirds of the times) are underlined.

| | SelectKBest (T-test) | | RFE (SVM) | | RFE (LR) | | RFE (RF) | |
|---|---|---|---|---|---|---|---|---|
| | feature | freq | feature | freq | feature | freq | feature | freq |
| 1 | *Curvature_mean_PD* | 26 | *Curvature_mean_PD* | 23 | *Curvature_mean_PD* | 28 | *Diameter_normalD2* | 25 |
| 2 | *CrossArea_mean_PD2* | 18 | *Curvature_integration_P* | 22 | *Eccentricity_std_P2* | 22 | *Curvature_mean_PD* | 19 |
| 3 | Curvature_integration_P | 13 | LineDist | 18 | Curvature_integration_P | 20 | Eccentricity_std_P2 | 15 |
| 4 | CrossArea_max_PD2 | 11 | Diameter_normalD2 | 18 | LineDist | 19 | Eccentricity_variation_D2 | 12 |
| 5 | Curvature_mean_P | 9 | Eccentricity_std_P2 | 17 | Ratio_Dpnordnor1 | 17 | Ratio_Dpnordnor1 | 10 |
| 6 | Ratio_Dpnordnor1 | 9 | PointDist | 6 | Diameter_normalD2 | 8 | Curvature_integration_P | 9 |

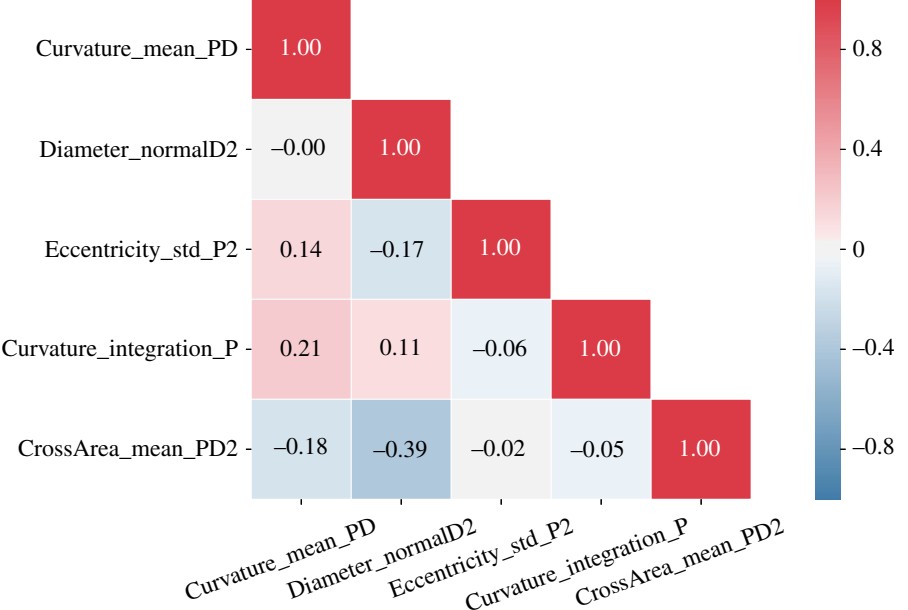

**Figure 3.** Pearson's correlation coefficient matrix for the selected five features.

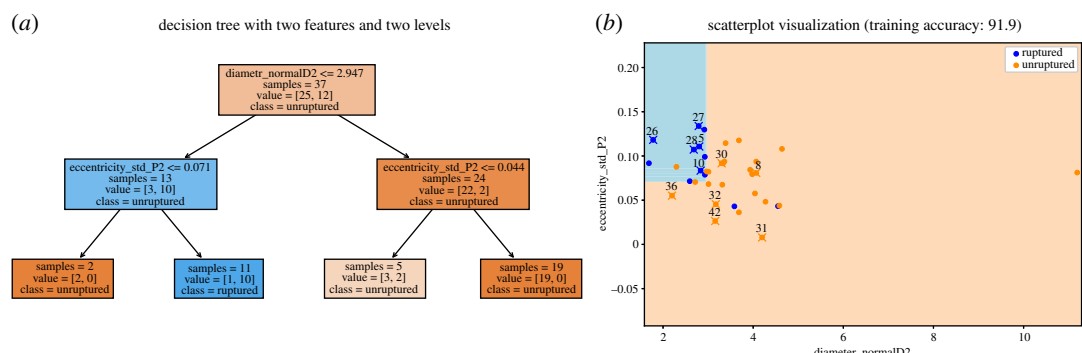

**Figure 4.** Decision tree fitted using only Diameter_normalD2 and Eccentricity_std_P2. The points with their case number labelled above are discussed in §3.3.1 (eccentricity) and/or §3.3.3 (diameter).

**Table 2.** Prediction performance of the feature sets, where $C$ = Curvature_mean_PD, $D$ = Diameter_normalD2 and $E$ = Eccentricity_std_P2.

| | [5 features] | | [4 features] | | [C + D + E] | | [C + D] | | [D + E] | | [C + E] | |
|---|---|---|---|---|---|---|---|---|---|---|---|---|
| | LOO | 80/20 | LOO | 80/20 | LOO | 80/20 | LOO | 80/20 | LOO | 80/20 | LOO | 80/20 |
| SVM (linear) | 94.6 | 89.4 | 86.5 | 88.0 | 86.5 | 83.6 | 83.8 | 81.8 | 78.4 | 72.2 | 81.1 | 77.6 |
| LR (l2 penalty) | 94.6 | 91.5 | 86.5 | 87.0 | 86.5 | 82.6 | 83.8 | 81.8 | 78.4 | 75.6 | 78.4 | 78.1 |
| KNN (k = 5) | 86.5 | 83.0 | 83.8 | 82.1 | 75.7 | 75.8 | 75.7 | 77.4 | 73.0 | 73.9 | 75.7 | 71.5 |
| decision tree | 89.2 | 82.1 | 89.2 | 82.2 | 83.8 | 78.4 | 73.0 | 76.0 | *83.8* | 79.2 | 75.7 | 66.4 |

Using this tree, the training accuracy is very high. We notice that curvature integration less than or equal to 0.523 only separates out one unruptured case. This may be unnecessary and could increase the risk of overfitting. We could remove the bottom four nodes and make it a tree with depth two. Finally, the two linear models, SVM and LR, have very similar performance with respect to the sign of coefficients. Table 3 shows the estimated coefficients using linear SVM and logistic regression.

**Table 3.** Coefficients estimated by linear SVM and LR.

| | [5 features] | | [4 features] | | [C + D + E] | | [C + D] | | [D + E] | | [C + E] | |
| --- | --- | --- | --- | --- | --- | --- | --- | --- | --- | --- | --- | --- |
| | SVM | LR | SVM | LR | SVM | LR | SVM | LR | SVM | LR | SVM | LR |
| Curvature_mean_PD | −0.889 | −1.182 | −0.891 | −1.29 | −1.227 | −0.611 | −1.229 | −0.608 | | | −0.867 | −1.228 |
| Diameter_normalD2 | −1.267 | −0.713 | −1.293 | −0.617 | −1.093 | −1.3 | −1.094 | −1.3 | −1.194 | −0.518 | | |
| Eccentricity_std_P2 | 0.247 | 0.175 | 0.237 | 0.213 | 0.147 | 0.216 | | | 0.248 | 0.198 | 0.362 | 0.12 |
| Curvature_integration_P | −1.311 | −1.063 | −1.229 | −0.993 | | | | | | | | |
| CrossArea_mean_PD2 | 0.217 | 0.48 | | | | | | | | | | |

In summary, our results indicate that higher rupture risk is associated with

— a smaller ratio of mean curvature between distal and proximal 1;
— a smaller diameter (unit mm) at distal 2 endpoint;
— a more variable eccentricity in proximal 2;
— a smaller integration of curvature in proximal; and
— a larger ratio of mean cross-sectional area between distal and proximal 2.

## 3.3. Feature exploration

To further explore the features selected by our method, we looked into ruptured and unruptured cases with significant difference in features such as eccentricity and curvature. We mainly focused on Eccentricity_std_P2 and Curvature_integration_P, since the other three features are more of a result from value ratios rather than an average of a geometric property over a specific segment. We plotted images of the corresponding segments below, along with the values of the variables related to these features along the segments, using arc length as the horizontal axis. Note that each case number corresponds to exactly one case, and the cases are not labelled consecutively from 1 to 37.

### 3.3.1. Eccentricity

The first feature we investigated was the standard deviation of eccentricity at segment Proximal2 (Eccentricity_std_P2). Cases 27 and 18, as well as Cases 26 and 36, are two pairs of (ruptured, unruptured) that were investigated initially. In figures 5 and 6, the visualization of segment Proximal2 is demonstrated for these cases, as well as the eccentricity at each point along segment Proximal2 against the arc length of segment. Figures labelled (c) plotted eccentricity for ruptured and unruptured case with mean eccentricity value of the specific case segment and error bars, and then together with both cases on the same scale of arc length. The mean eccentricity is constant for a specific segment of a specific case, but differs across segments and cases. The error bars place each point at the centre of the vertical bar, with lengths of each error bar above and below the data points determined by how far each eccentricity point deviates from the mean eccentricity.

As observed from figures, the standard deviation of eccentricity at segment Proximal2 for ruptured cases is generally larger than that for unruptured cases. The trend follows how one of our classification methods, decision trees, predicts the risk of rupture based on the dataset. As figure 7 indicates, Eccentricity_std_P2 plays an important role in classifying the cases, and cases with Eccentricity_std_P2 smaller than the threshold are mostly classified as unruptured. Such trend also makes intuitive sense. Eccentricity measures how elliptical a two-dimensional shape is, and in this context it measures the cross-sectional surface at a specific point in the blood vessel. A great variation in eccentricity at segment Proximal2, therefore, indicates possible dramatic changes in the shape of blood vessel upstream of the aneurysm, which would in turn influence the blood flow and facilitate the rupture.

More pairs of (ruptured, unruptured) cases with significant difference in Eccentricity_std_P2 are shown in electronic supplementary material, figures A.5 and A.6. These cases further demonstrate visually that unruptured cases are more likely to have smaller standard deviation in eccentricity at segment Proximal2.

*Remark:* A potential concern is the difference in segment arc length, which may potentially introduce bias to the results since the length of segment Proximal2 varies in different cases due to the criterion each segment is defined. To address this issue, we have computed the value of Eccentricity_std_P2 by reducing the segment length and found that it is still greater for ruptured case, when the same segment length is used for both ruptured and unruptured cases. In addition, the two pairs (Case 27, Case 42) and (Case 26, Case 36) demonstrate two different kinds of relations between segments for ruptured and unruptured cases: Case 27 (ruptured) is longer in arc length than Case 18 (unruptured), whereas Case 26 (ruptured) is much shorter in arc length than Case 36 (unruptured). This implies that there are no certain relations between segments for ruptured and unruptured cases.

### 3.3.2. Curvature

The integral of curvature over segment Proximal1 (Curvature_integration_P is another important feature for our classification method. The integral is calculated using the method of trapezoidal numerical integration in Matlab, and each segment has a corresponding integral of curvature over arc length (details of formula,

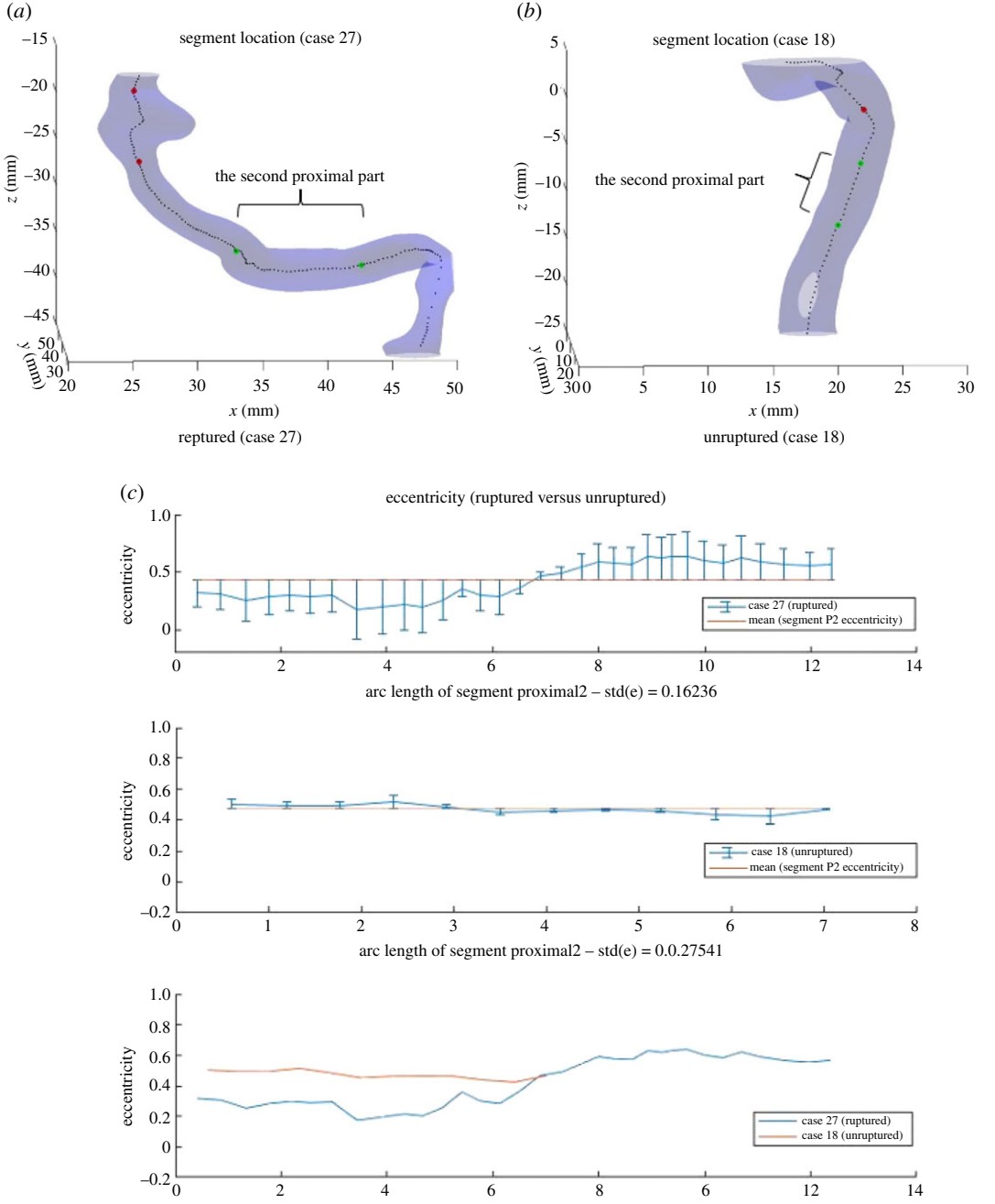

**Figure 5.** Segments Proximal2 for Case 27 (ruptured) and Case 18 (unruptured), and eccentricity at each point along the segment.

see electronic supplementary material, appendix A.4.4). Similar to the investigation of Eccentricity_std_P2, (ruptured, unruptured) pairs with significant difference in Curvature_integration_P are chosen, and curvature is plotted against arc length of segment Proximal1 in figure 8 (more examples are given in electronic supplementary material, figures A.7 and A.8 in appendix). Based on the these figures, we can make the following observations.

— Curvature at segment Proximal1 for ruptured cases tends to have a wave-like pattern where values of curvature vary more, and have larger maximum and lower minimum than that for unruptured cases. The integral of the curve, therefore, partially captures such trend. Even though other mathematical measures such as standard deviation might intuitively be more indicative of how curvature varies, such features did not come up during our feature extraction method.

— Curvature_integration_P also serves to distinguish cases where Eccentricity_std_P2 fails. In the (ruptured, unruptured) pair (Case 2, Case 45), the distinguishing between Eccentricity_std_P2 is not evident visually.

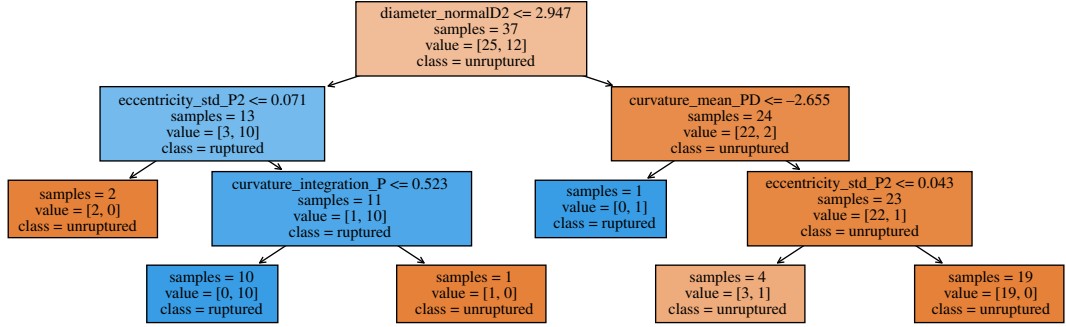

**Figure 6.** Segments Proximal2 for Case 26 (ruptured) and Case 36 (unruptured), and eccentricity at each point along the segment.

decision tree with four features and three levels (training accuracy: 97.3)

**Figure 7.** Decision tree fitted using Curvature_mean_PD, Diameter_normalD2, Eccentricity_std_P2 and Curvature_integration_P.

With curvature, however, it is much more evident that the ruptured case has a higher maximum, lower minimum, and larger extent of variation. In such cases, Curvature_integration_P complements Eccentricity_std_P2 and helps predict the risk of rupture.

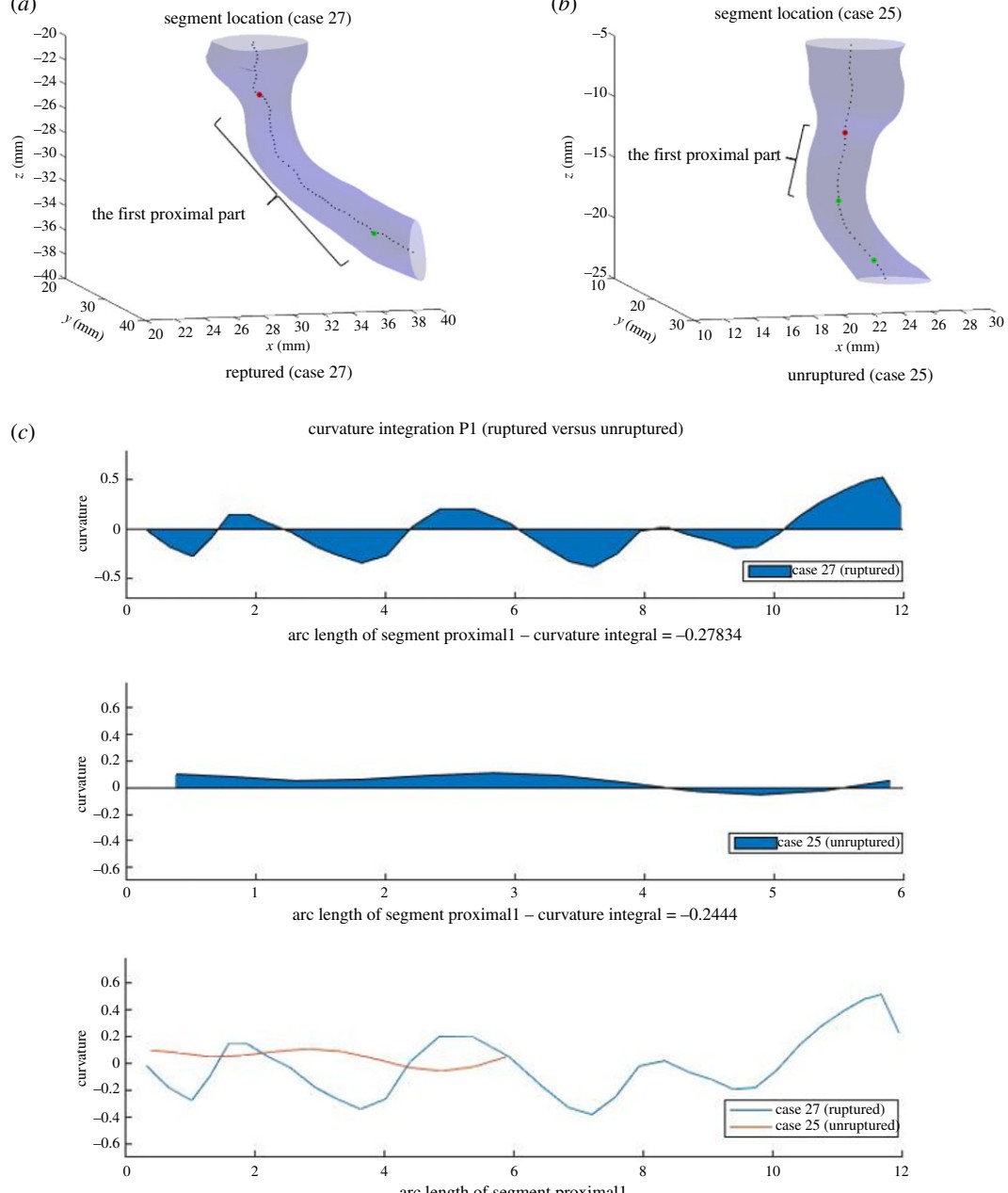

**Figure 8.** Segments Proximal1 for Case 27 (ruptured) and Case 25 (unruptured), and curvature at each point along the segment, as well as the integral of curvature.

### 3.3.3. Diameter

In addition to eccentricity and curvature, we also briefly investigated Diameter_normal_D2, which is the diameter of the blood vessel cross section at the point of segmentation D2. Diameter is calculated by the major axis length of blood vessel cross sections. Different from Eccentricity_std_P2 and Curvature_integration_P, which are derived by measuring the entire segment of interest, Diameter_normal_D2 is a point value. Therefore, we looked into diameters over the two distal segments, D1 and D2, to further explore this feature.

As figure 9 indicates, ruptured cases tend to have smaller diameters than unruptured cases over the two distal segments D1 and D2. More example can be found in electronic supplementary material, figure A.9 in appendix. However, the relation between diameters each at segment D1 and D2 is not clear: segment D1 can have larger or smaller diameters than segment D2 for both ruptured and unruptured cases. This implies that the diameter varies. Taking the point value at points of segmentation is probably why diameter at segment D1 is not picked. Also, features relating to the

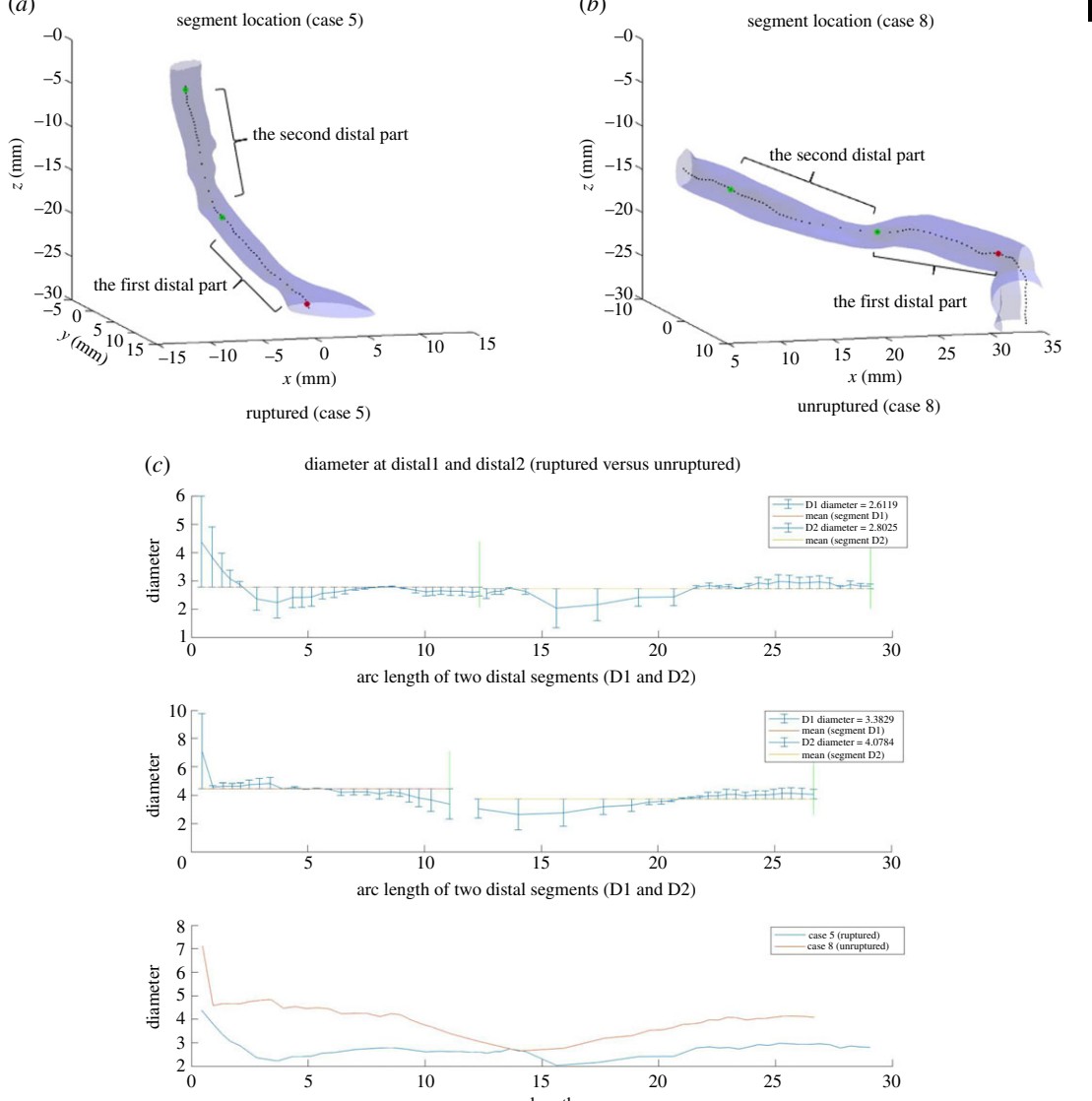

**Figure 9.** Segments Distal1 and Distal2 for Cases 5 and 8, and diameter at each point along the two segments.

max value, mean, standard deviation, integral and variation of diameters at each distal segment were included in the dataset but did not come up during feature selection.

### 3.3.4. Combination of features

Our previous examples demonstrated that eccentricity and curvature are two useful features for predicting rupture risk. However, there exist cases that it is not sufficient to make a prediction by relying on only one or both of these two features. Below we present three examples where (i) Eccentricity_std_P2 alone fails to make a prediction and adding Curvature_integration_P is sufficient; (ii) using both features also fails to separate the cases, and other features have to be used; and (iii) a rare case that even with all five features, it is still not possible to make a prediction.

*Two features*. Figure 10 demonstrates an example where Eccentricity_std_P2 fails to separate ruptured and unruptured cases, while curvature gives a clear distinction between the two cases.

*Two + Features*. Figure 11 demonstrates an example where both eccentricity and curvature fails to follow the trend we have discussed before. In this specific scenario, Case 2 (ruptured) and Case 23 (unruptured) can be separated by the feature Curvature_mean_PD, according to our model by Decision Tree.

*Misclassified cases*. Finally, there are misclassified cases where none of the features correctly predicted the risk of rupture. The pair (Case 1, Case 32) is an example where eccentricity and curvature are not

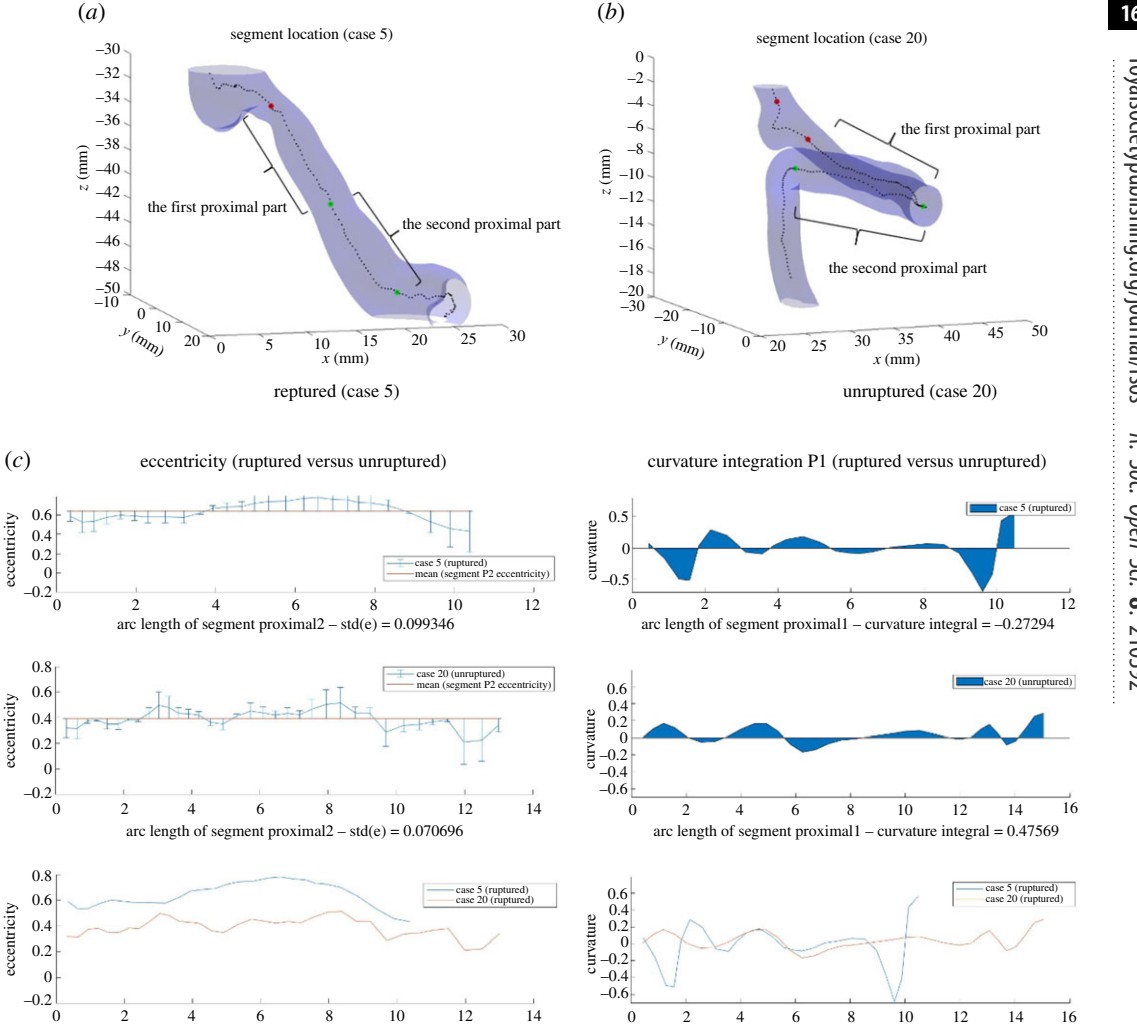

**Figure 10.** Segments Proximal1 and Proximal2 for Cases 5 and 20, and eccentricity and curvature, respectively, at each point along segment Proximal2/Proximal1. The difference in Eccentricity_std_P2 is not evident, whereas the difference in the variation of curvature at segment Proximal1 is more clear.

sufficient, and including additional features also fails in distinguishing the two cases by the Decision Tree model, and Case 1 is misclassified as shown in figure 12.

# 4. Discussion and conclusion

In this paper, we extended our previous study on predicting the risk of rupture of brain aneurysms using a geometric-based approach [5]. We have significantly improved our previous results in several aspects. First of all, we have improved the accuracy by more than 10% with five selected features. Secondly, we have excluded features related to aneurysm itself and instead focused on geometry of the vasculature on both side of the aneurysm. This is important as it provides additional information to the doctors when they try to decide whether operation is required, which itself also involves significant risks. Thirdly, we also reduced the number of features needed for the prediction, therefore greatly increased the interpretability, which is a common problem of machine learning algorithms, and reduced the risk of overfitting when the sample size is small, which is also common in medicine.

Specifically, using the new approach of vessel segmentation and the aforementioned models, having five segments arguably achieves the best balance. With 'Diameter_normalD2', 'Curvature_mean_PD', 'Eccentricity_std_P2', 'Curvature_integration_P' and 'CrossArea_mean_PD2', we can achieve 94.6% accuracy on the given dataset using leave-one-out cross validation. A reasonable accuracy, 86.5% or 83.8%, could also be achieved with a smaller feature set of three or two features.

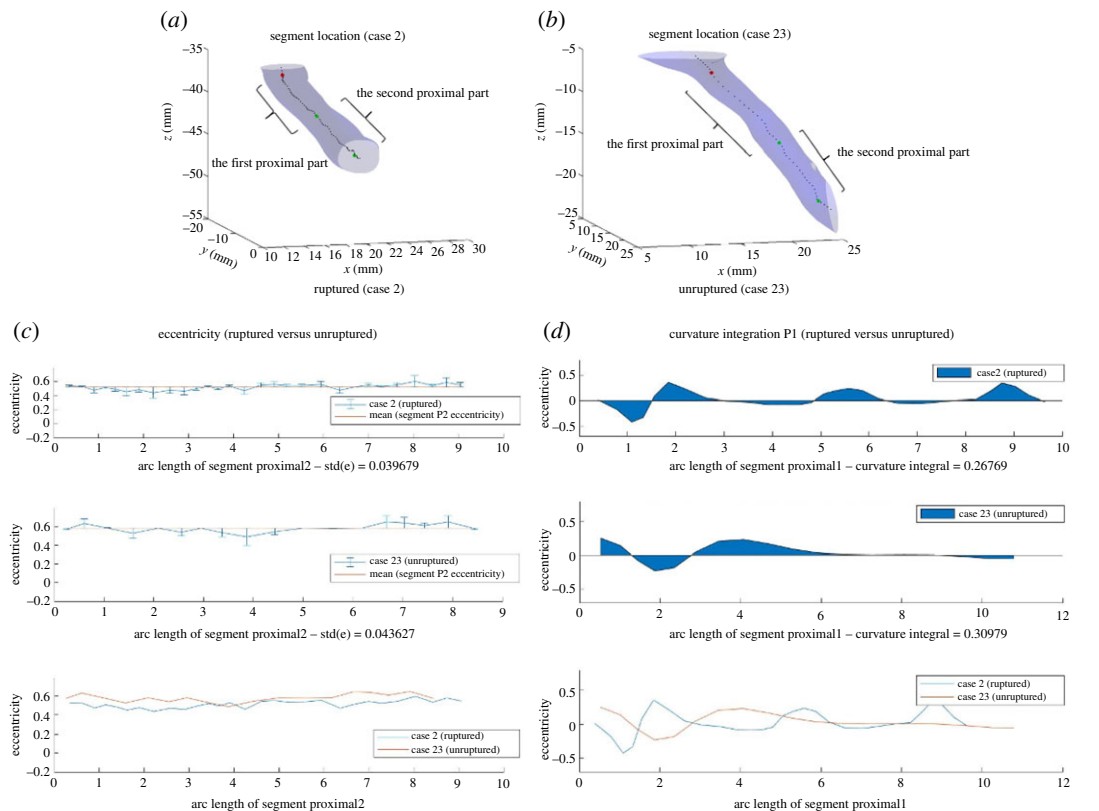

**Figure 11.** Segments Proximal1 and Proximal2 for Cases 2 and 23, and eccentricity and curvature respectively at each point along segment Proximal2/Proximal1. Neither eccentricity nor curvature follows the trend we have discussed previously. By Decision Tree, Curvature_mean_PD separates the cases.

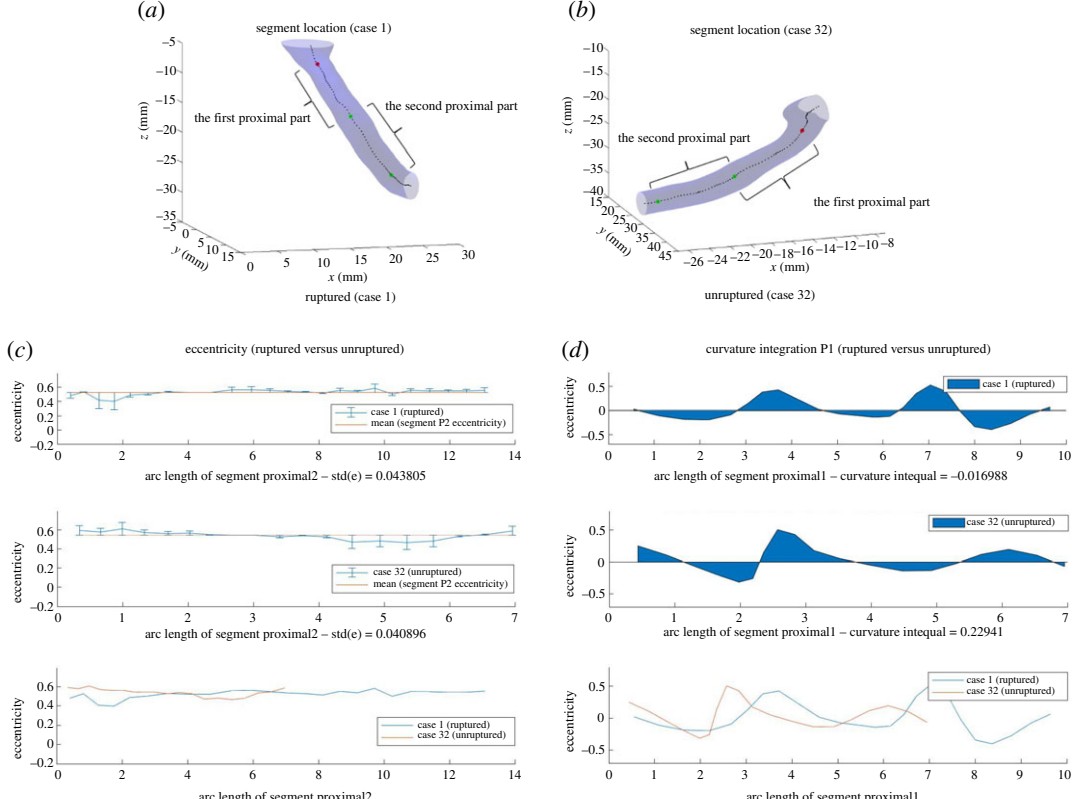

**Figure 12.** Segments Proximal1 and Proximal2 for Cases 1 and 32, and eccentricity and curvature respectively at each point along segment Proximal2/Proximal1 for (Case 1, Case 32). Neither eccentricity nor curvature follows the trend we have discussed previously. Case 1 is misclassified by Decision Tree model.

**Table 4.** Selected features with different threshold.

| threshold | std | | 0.9 std | | 1.1 std | |
| --- | --- | --- | --- | --- | --- | --- |
| criteria | top2 | ≥20 | top2 | ≥20 | top2 | ≥20 |
| Curvature_mean_PD | x | x | x | x | x | x |
| Diameter_normalD2 | x | x | x | x | x | x |
| Eccentricity_std_P2 | x | x | | | x | x |
| Curvature_integration_P | x | x | x | x | | |
| CrossArea_mean_PD2 | x | | x | x | x | |
| Solidity_integration_PD2 | | | | | x | x |
| MajorAxis_mean_D2 | | | x | x | | |

**Table 5.** Best accuracy achievable using a set of two features.

| threshold | std | | 0.9 std | | 1.1 std | |
| --- | --- | --- | --- | --- | --- | --- |
| CV | LOO | 80/20 | LOO | 80/20 | LOO | 80/20 |
| SVM (linear) | 83.8 | 81.8 | 75.7 | 77.1 | 86.5 | 84.5 |
| logistic regression | 83.8 | 81.8 | 81.1 | 80.5 | 89.2 | 85.6 |

In the previous study [5], authors followed the clinic doctors' common intentions that aneurysm-specific features should play important roles on the rupture risk. However, when we used the aneurysm-specific features, we discovered that four out of the five top features involve characteristics of surrounding blood vessels. Consequently, we hypothesized that parent blood vessels may play more important roles on rupture and can be used for rupture prediction independently. In fact, if we add the aneurysm-specific features to the features in our new dataset, only two aneurysm-specific features—'Solidity_integration' and 'Eccentricity_integration'—pass the T-test with a 0.1 threshold. Using the combined dataset, the top 2 features from each of the four feature selection algorithms described in the paper are as follows:

— SelectKBest: Curvature_mean_PD, CrossArea_mean_PD2
— RFE(SVM): Curvature_mean_PD, Curvature_integration_P
— RFE(LR): Curvature_mean_PD, Eccentricity_std_P2
— RFE(RF): Diameter_normalD2, Curvature_mean_PD

This ranking list is exactly the same as the ranking list using the non-aneurysm-specific dataset. This result confirms our hypothesis that geometry of parent blood vessels is also quite an important factor for fusiform aneurysms rupture. This is also consistent with previous studies on haemodynamics of aneurysms that the parent vessel of the aneurysm is strongly associated with haemodynamics characteristics which in turn contributes to aneurysm formation, growth and rupture [22].

One concern about the model is the uncertainty qualification during the feature generation induced by the segmentation threshold (std of cross-sectional area of each entire vessel). To address this problem, we introduce a variation of this threshold by ±10%. As described in the paper, we considered two methods for selecting top features: one is to take the union set of the top 2 features of each selection algorithm, the other is to take the set of features with a frequency of at least 20 (out of 30) times. Table 4 compares the top features selected under different segmentation thresholds. It is shown that most of selected features are the same between the sets using the different thresholds, supporting stability of the algorithm.

In addition, table 5 compares the best accuracy achievable using a set of two features, specifically Curvature_mean_PD and Diameter_normalD2. The method we are using is robust since changing the parameter (threshold) has small effects as shown by our numerical experiments.

The main limitation of our result is that with only 37 cases, there is a significant risk of overfitting. Some measures, such as limiting the feature set size and using only subsets as the input to feature selection models, have been taken to partially address this issue. As more data become available, we

plan to address this issue by perform validation using new cases as well as other approaches such as neural networks in future studies.

The occurrence and rupture of cerebral aneurysms are the result of multiple factors. Therefore, a truly individualized assessment of the risk of aneurysm rupture must also be based on multiple dimensions (genetics, proteomics, haemodynamics, etc.) [22]. However, such multi-dimensional and multi-factor research is difficult to implement, and the cost is huge. Although it is not accurate to predict the rupture risk of an aneurysm or the prognosis of treatment based on a single factor in clinical practice, it is still beneficial to consider the geometric parameters of the blood vessel that are very easy to obtain in the process of treatment. It is possible to conduct independent research based on a single factor and assign different weights to data from different dimensions in subsequent research.

At the same time, our results confirmed that the parent vessel plays an important role in fusiform aneurysm rupture. We are prospectively collecting the contralateral normal artery data for an autologous control study. In the future, we will generalize current work and develop a model to predict aneurysm formation risk by comparing the blood vessel structures for aneurysm patients as well as for healthy subjects.

In addition, many studies have found that changing the diameter [23], angle [24,25] and other geometric morphological characteristics of the parent artery can promote the healing of the aneurysm. Since the acquisition of geometric morphological parameters is simple, a more in-depth study of the geometric morphology of the parent artery is expected to guide clinicians in the process of treating cerebral aneurysms in real-time judgement of the rationality of the treatment strategy and the treatment effect.

Ethics. Ethical approval for the retrospective study was obtained from the institutional review board of Changhai Hospital, Shanghai, China. The requirement for informed consent was waived by the review board due to the retrospective design of the study.

Data accessibility. Data and relevant code for this research work are stored in GitHub: https://github.com/RPTS/Aneurysm_2020_Code, and have been archived within the Zenodo repository: http://doi.org/10.5281/zenodo.4586810.

Authors' contributions. R.P. carried out the image and statistical analysis, participated in the design of the study and drafted the manuscript; S.L. participated in statistical analysis and helped draft the manuscript; Y.F. collected image data, participated in image analysis and study design, and helped draft the manuscript; S.X. participated in image analysis and commented on the manuscript; A.G. and H.H. designed the study, coordinated the study, and commented on the manuscript. All authors gave and approval for publication.

Competing interests. We declare we have no competing interests.

Funding. This research was supported in part by the Fields Institute (S.L., R.P. and H.H.), NSERC (H.H.), and startup funds from Duke Kunshan University, NSFC (12071190) (S.X.).

Acknowledgements. The authors wish to thank Dr Xiukun Zhao for her valuable insights during the course of this project. Authors also would like to thank the Fields Institute for hosting the summer research program and the Fields CQAM lab on health analytics and multidisciplinary modeling for providing the support to Shixuan Li and Ruiqi Pan.

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
