## [Peer Review File · Royal Society Open Science]

Review History

RSOS-201801.R0 (Original submission)

Review form: Reviewer 1

Is the manuscript scientifically sound in its present form?

Yes

Are the interpretations and conclusions justified by the results?

Yes

Is the language acceptable?

Yes

Do you have any ethical concerns with this paper?

No

Have you any concerns about statistical analyses in this paper?

No

Recommendation?

Major revision is needed (please make suggestions in comments)

Comments to the Author(s)

In this study, Li et al studied the geometric features of vertebral aneurysm for predicting the risk of its rupture. They first extracted 382 non-aneurysm-related features at the distal and proximal parts of the blood vessel, and then the feature selection, and finally the classification using 4 different well-developed ML models. This study tried to address a challenging and yet unsolved research question on predicting rupture risk through geometrical features. While because of a lack of samples, only 37 subjects are involved in this study, the applicability of this would require a large cohort of patient data. This study is closely related to the previous study by Zhao et al. My major concerns are

(1) In their previous study, Zhao et al, studied 5 geometric features of VAFA morphology and achieved state-of-the-art classification performance. It is necessary to compare the geometric features proposed in this study, what are the differences, and what about combining the two sets of features together? My suggestion is to conduct a thorough study by including both the geometric features from their previous study and the features beyond aneurysm proposed in this study, then it would address the geometrical features of VAFA morphology properly.

(2) It is necessary to share the codes for reproducibility. I would also suggest sharing the 3D geometrical data if they can so that it will benefit the whole community.

(3) Line 14, they mentioned 'a new segmentation method', the authors can make it clear it is not about image segmentation.

(4) Section 2. about the segmentation, it is not clear to me how they use cross-sectional area first-order difference, curvature, and torsion as potential criteria for re-segmentation. Detailed procedures and quantitative criteria shall be provided. In the followed paragraph, the authors have tried to explain but seems very human-dependent. The authors can also perform some uncertainty quantification on the thresholds and related parameters, and also inter-observer and intra-observer variability. The reliability and reproducibility of the segmentation would be very essential for the geometrical feature extraction.

(5) Section 2.4, please explain how the integration is done mathematically along the curve. It is not clear 'The ratio between values, along with the angle and distance between two centerline tangent vectors at the neck of the bulge', considering providing a schematic illustration

(6) section 2.6, more details can be added and discussed. For example, in SVM, the reason of choosing a linear kernel, what is the performance using a nonlinear kernel, similar for the logistic regression. In the K-Nearest Neighbours, studying different k will be helpful for choosing the k value. What is the majority vote? The choice of random tree depth needs more analysis.

(7) please add the implementation details of the feature selection and modeling

(8) In the results, the authors mentioned the 3 segments and 7 segments. An illustration of how those segments are determined will be helpful

(9) For Table 1, I would suggest the authors stack the feature ranking together using a bar chart, so that it is much easier to spot the importance of each feature

(10) page 7, line 6, what is the unit for the diameter?

(11) Discussions are more like results. Please merge with the result section, only discussing their implications in the discussion section, including uncertainties, etc.

(12) Figure 5, please add units for the horizontal and vertical axis. The total cases are 37, then what is case 42? What is the meaning of the error bar in (c), is the mean eccentricity constant? Figure (c) is small and hard to see. Figures 6, 7, and 8 can be placed into the appendix.

(13) Figure 9 has a similar issue as Figure 5. Please define the integral of curvature mathematically, it is not clear at (c) how the integral varies along the arc length. Figures 10 and 11 can be placed in the Appendix.

(14) Figure 12, similar issue. Figure 13 can be included in the appendix

(15) It would be helpful if the authors can include the definitions of some geometric indices into the main text, rather than in the appendix, to make the paper self-contained. For example, the curvature and its integration, etc. The Figure in the appendix should be renumbered, not following the main text. A.2 should be included in the main text for the convenience of reading. Figure 19(b) is not exactly the convex hull of the blue curve.

Review form: Reviewer 2

Is the manuscript scientifically sound in its present form?

No

Are the interpretations and conclusions justified by the results?

Yes

Is the language acceptable?

No

Do you have any ethical concerns with this paper?

No

Have you any concerns about statistical analyses in this paper?

No

Recommendation?

Reject

Comments to the Author(s)

Overview

The present manuscript is devoted to the investigation of various morphological features to predict rupture. The same 37 fusiform cases are studied as already published by the present authors in 2018 (Zhao et al., 2018 Vertebral artery fusiform aneurysm geometry in predicting rupture risk. Royal Society Open Science 5(10), 180780. (doi:10.1098/rsos.180780)). The present manuscript is incremental compared to this previous work by the authors. Therefore, I do not recommend the publication of the present manuscript in Royal Society Open Science.

Major comments:

Further details regarding the segmentation should be provided, including methods and smoothing. Some of the investigated morphological features are highly sensitive on these segmentation steps.

The authors state that the blood flow essential for the rupture prediction of brain aneurysm. No flow simulations are involved in the present study, only morphological parameters are investigated. It is not clear how the distal parts influence the blood flow (Figures 12 & 13).

The number of the investigated cases is relatively low to obtain representative results.

Minor comments:

British (e.g. modelling, centre) and US spelling (e.g. categorized, organized, modeling, optimized, center) are mixed up. Only one of the spelling should be used in the manuscript.

Page 4, line 42: the -> The

Page 13, line 26: Aneurysms -> aneurysms

Decision letter (RSOS-201801.R0)

Dear Dr Xu

The Editors assigned to your paper RSOS-201801 "Predicting the Risk of Rupture for Vertebral Aneurysm based on Geometric Features of Blood Vessels" have made a decision based on their reading of the paper and any comments received from reviewers.

Regrettably, in view of the reports received, the manuscript has been rejected in its current form. However, a new manuscript may be submitted which takes into consideration these comments.

We invite you to respond to the comments supplied below and prepare a resubmission of your manuscript. Below the referees' and Editors' comments (where applicable) we provide additional requirements. We provide guidance below to help you prepare your revision.

Please note that resubmitting your manuscript does not guarantee eventual acceptance, and we do not generally allow multiple rounds of revision and resubmission, so we urge you to make every effort to fully address all of the comments at this stage. If deemed necessary by the Editors, your manuscript will be sent back to one or more of the original reviewers for assessment. If the original reviewers are not available, we may invite new reviewers.

Please resubmit your revised manuscript and required files (see below) no later than 06-Jul-2021. Note: the ScholarOne system will 'lock' if resubmission is attempted on or after this deadline. If you do not think you will be able to meet this deadline, please contact the editorial office immediately.

Please note article processing charges apply to papers accepted for publication in Royal Society Open Science (<https://royalsocietypublishing.org/rsos/charges>). Charges will also apply to papers transferred to the journal from other Royal Society Publishing journals, as well as papers submitted as part of our collaboration with the Royal Society of Chemistry (<https://royalsocietypublishing.org/rsos/chemistry>). Fee waivers are available but must be requested when you submit your manuscript (<https://royalsocietypublishing.org/rsos/waivers>).

Thank you for submitting your manuscript to Royal Society Open Science and we look forward to receiving your resubmission. If you have any questions at all, please do not hesitate to get in touch.

on behalf of Dr Peter Stewart (Associate Editor) and Mark Chaplain (Subject Editor)
openscience@royalsociety.org

Associate Editor Comments to Author (Dr Peter Stewart):

Associate Editor: 1

Comments to the Author:

We have now received two referee reports for this paper. Based on these detailed reviews we have decided to reject the paper in its current form.

Reviewer comments to Author:

Reviewer: 1

Comments to the Author(s)

In this study, Li et al studied the geometric features of vertebral aneurysm for predicting the risk of its rupture. They first extracted 382 non-aneurysm-related features at the distal and proximal parts of the blood vessel, and then the feature selection, and finally the classification using 4 different well-developed ML models. This study tried to address a challenging and yet unsolved research question on predicting rupture risk through geometrical features. While because of a lack of samples, only 37 subjects are involved in this study, the applicability of this would require a large cohort of patient data. This study is closely related to the previous study by Zhao et al. My major concerns are

(1) In their previous study, Zhao et al, studied 5 geometric features of VAFA morphology and achieved state-of-the-art classification performance. It is necessary to compare the geometric features proposed in this study, what are the differences, and what about combining the two sets of features together? My suggestion is to conduct a thorough study by including both the geometric features from their previous study and the features beyond aneurysm proposed in this study, then it would address the geometrical features of VAFA morphology properly.

(2) It is necessary to share the codes for reproducibility. I would also suggest sharing the 3D geometrical data if they can so that it will benefit the whole community.

(3) Line 14, they mentioned 'a new segmentation method', the authors can make it clear it is not about image segmentation.

(4) Section 2. about the segmentation, it is not clear to me how they use cross-sectional area first-order difference, curvature, and torsion as potential criteria for re-segmentation. Detailed procedures and quantitative criteria shall be provided. In the followed paragraph, the authors have tried to explain but seems very human-dependent. The authors can also perform some uncertainty quantification on the thresholds and related parameters, and also inter-observer and intro-observer variability. The reliability and reproducibility of the segmentation would be very essential for the geometrical feature extraction.

(5) Section 2.4, please explain how the integration is done mathematically along the curve. It is not clear 'The ratio between values, along with the angle and distance between two centerline tangent vectors at the neck of the bulge', considering providing a schematic illustration

(6) section 2.6, more details can be added and discussed. For example, in SVM, the reason of choosing a linear kernel, what is the performance using a nonlinear kernel, similar for the logistic regression. In the K-Nearest Neighbours, studying different k will be helpful for choosing the k value. What is the majority vote? The choice of random tree depth needs more analysis.

(7) please add the implementation details of the feature selection and modeling

(8) In the results, the authors mentioned the 3 segments and 7 segments. An illustration of how those segments are determined will be helpful

(9) For Table 1, I would suggest the authors stack the feature ranking together using a bar chart, so that it is much easier to spot the importance of each feature

(10) page 7, line 6, what is the unit for the diameter?

(11) Discussions are more like results. Please merge with the result section, only discussing their implications in the discussion section, including uncertainties, etc.

(12) Figure 5, please add units for the horizontal and vertical axis. The total cases are 37, then what is case 42? What is the meaning of the error bar in (c), is the mean eccentricity constant? Figure (c) is small and hard to see. Figures 6, 7, and 8 can be placed into the appendix.

(13) Figure 9 has a similar issue as Figure 5. Please define the integral of curvature mathematically, it is not clear at (c) how the integral varies along the arc length. Figures 10 and 11 can be placed in the Appendix.

(14) Figure 12, similar issue. Figure 13 can be included in the appendix

(15) It would be helpful if the authors can include the definitions of some geometric indices into the main text, rather than in the appendix, to make the paper self-contained. For example, the curvature and its integration, etc. The Figure in the appendix should be renumbered, not following the main text. A.2 should be included in the main text for the convenience of reading. Figure 19(b) is not exactly the convex hull of the blue curve.

Reviewer: 2

Comments to the Author(s)

Overview

The present manuscript is devoted to the investigation of various morphological features to predict rupture. The same 37 fusiform cases are studied as already published by the present

authors in 2018 (Zhao et al., 2018 Vertebral artery fusiform aneurysm geometry in predicting rupture risk. Royal Society Open Science 5(10), 180780. (doi:10.1098/rsos.180780)). The present manuscript is incremental compared to this previous work by the authors. Therefore, I do not recommend the publication of the present manuscript in Royal Society Open Science.

Major comments:

Further details regarding the segmentation should be provided, including methods and smoothing. Some of the investigated morphological features are highly sensitive on these segmentation steps.

The authors state that the blood flow essential for the rupture prediction of brain aneurysm. No flow simulations are involved in the present study, only morphological parameters are investigated. It is not clear how the distal parts influence the blood flow (Figures 12 & 13).

The number of the investigated cases is relatively low to obtain representative results.

Minor comments:

British (e.g. modelling, centre) and US spelling (e.g. categorized, organized, modeling, optimized, center) are mixed up. Only one of the spelling should be used in the manuscript.

Page 4, line 42: the -> The

Page 13, line 26: Aneurysms -> aneurysms

===PREPARING YOUR MANUSCRIPT===

If you have been asked to revise the written English in your submission as a condition of publication, you must do so, and you are expected to provide evidence that you have received language editing support. The journal would prefer that you use a professional language editing service and provide a certificate of editing, but a signed letter from a colleague who is a native speaker of English is acceptable. Note the journal has arranged a number of discounts for authors

using professional language editing services
(<https://royalsociety.org/journals/authors/benefits/language-editing/>).

===PREPARING YOUR REVISION IN SCHOLARONE===

<https://royalsociety.org/journals/authors/author-guidelines/#supplementary-material> to include a suitable title and informative caption. An example of appropriate titling and captioning may be found at https://figshare.com/articles/Table_S2_from_Is_there_a_trade-

off_between_peak_performance_and_performance_breadth_across_temperatures_for_aerobic_sc
ope_in_teleost_fishes_/3843624.

Author's Response to Decision Letter for (RSOS-201801.R0)

See Appendix A.

RSOS-210392.R0

Review form: Reviewer 1

Is the manuscript scientifically sound in its present form?

No

Are the interpretations and conclusions justified by the results?

Yes

Is the language acceptable?

Yes

Do you have any ethical concerns with this paper?

No

Have you any concerns about statistical analyses in this paper?

No

Recommendation?

Major revision is needed (please make suggestions in comments)

Comments to the Author(s)

The authors have tried to address all my comments, thanks for their efforts. While some of the comments are not fully resolved, thus I do not recommend for publication of the current revision. They are

(1) Original comment 1: I am not convinced that the aneurysm-specific features can be ignored. It is necessary to have a comparison among these three combinations: (1) only aneurysm-specific features, (2) only use the non-aneurysm-specific features, and (3) the combinations of the above two features. This will be interesting to the readers and also make the study completed, rather than incremental and somehow disconnected from their previous study.

(2) Original Comment 4 about uncertainty quantification: Again this is important for the classification. If the uncertainty from the feature extraction is significantly large, then the classification based on those features could be problematic. The authors seem to have not addressed or discussed this concern in the revision.

Review form: Reviewer 2

Is the manuscript scientifically sound in its present form?

Yes

Are the interpretations and conclusions justified by the results?

No

Is the language acceptable?

Yes

Do you have any ethical concerns with this paper?

No

Have you any concerns about statistical analyses in this paper?

No

Recommendation?

Accept with minor revision (please list in comments)

Comments to the Author(s)

Major comment:

The authors have virtually removed the aneurysms in the investigated cases and they try to find characteristic features of the vessel geometries in order to predict the risk of the aneurysm rupture. This is certainly interesting; however the blood vessel and the features of the aneurysms should be treated simultaneously.

Have the authors compared the cases after removing the aneurysms with healthy subjects? It would be interesting to see the blood vessel structures for aneurysm patients as well as for healthy subjects. The presented tools might predict the risk of developing an aneurysm for certain blood vessels.

The following statement should be refined "The geometric properties of the proximal vessels could potentially impact the normal blood flow, and those of the distal vessels may reflect such change upstream." (Section 2.3)

The question is not whether the flow is normal or not. It is more important to realize that the blood flow is highly influenced by the vessel geometry, e.g., the inflow-jet in the aneurysm is absolutely determined by the vessel.

Unfortunately, these hemodynamics effects are missing from the present study.

Decision letter (RSOS-210392.R0)

Dear Dr Xu

The Editors assigned to your paper RSOS-210392 "Predicting the Risk of Rupture for Vertebral Aneurysm based on Geometric Features of Blood Vessels" have now received comments from reviewers and would like you to revise the paper in accordance with the reviewer comments and any comments from the Editors. Please note this decision does not guarantee eventual acceptance.

Please submit your revised manuscript and required files (see below) no later than 21 days from today's (ie 12-May-2021) date. Note: the ScholarOne system will 'lock' if submission of the revision is attempted 21 or more days after the deadline. If you do not think you will be able to meet this deadline please contact the editorial office immediately.

on behalf of Dr Peter Stewart (Associate Editor) and Mark Chaplain (Subject Editor)
openscience@royalsociety.org

Associate Editor Comments to Author (Dr Peter Stewart):
Comments to the Author:

The reports have now been received from the referees. Both referees note that the paper has improved, but both complain loudly that the aneurysm specific features cannot be ignored, given the importance attributed to them in your previous study. This point should be addressed in the revision. As referee 1 suggests, it would be good to have a comparison of the three combinations: (1) only aneurysm-specific features, (2) only non-aneurysm-specific features, and (3) the combination of the two.

Reviewer comments to Author:

Reviewer: 1

Comments to the Author(s)

The authors have tried to address all my comments, thanks for their efforts. While some of the comments are not fully resolved, thus I do not recommend for publication of the current revision. They are

(1) Original comment 1: I am not convinced that the aneurysm-specific features can be ignored. It is necessary to have a comparison among these three combinations: (1) only aneurysm-specific features, (2) only use the non-aneurysm-specific features, and (3) the combinations of the above two features. This will be interesting to the readers and also make the study completed, rather than incremental and somehow disconnected from their previous study.

(2) Original Comment 4 about uncertainty quantification: Again this is important for the classification. If the uncertainty from the feature extraction is significantly large, then the classification based on those features could be problematic. The authors seem to have not addressed or discussed this concern in the revision.

Reviewer: 2

Comments to the Author(s)

Major comment:

The authors have virtually removed the aneurysms in the investigated cases and they try to find characteristic features of the vessel geometries in order to predict the risk of the aneurysm rupture. This is certainly interesting; however the blood vessel and the features of the aneurysms should be treated simultaneously.

Have the authors compared the cases after removing the aneurysms with healthy subjects? It would be interesting to see the blood vessel structures for aneurysm patients as well as for healthy subjects. The presented tools might predict the risk of developing an aneurysm for certain blood vessels.

The following statement should be refined "The geometric properties of the proximal vessels could potentially impact the normal blood flow, and those of the distal vessels may reflect such change upstream." (Section 2.3)

The question is not whether the flow is normal or not. It is more important to realize that the blood flow is highly influenced by the vessel geometry, e.g., the inflow-jet in the aneurysm is absolutely determined by the vessel.

Unfortunately, these hemodynamics effects are missing from the present study.

===PREPARING YOUR MANUSCRIPT===

Please ensure that you include an acknowledgements' section before your reference list/bibliography. This should acknowledge anyone who assisted with your work, but does not

qualify as an author per the guidelines at <https://royalsociety.org/journals/ethics-policies/openness/>.

===PREPARING YOUR REVISION IN SCHOLARONE===

- Ensure that your data access statement meets the requirements at <https://royalsociety.org/journals/authors/author-guidelines/#data>. You should ensure that you cite the dataset in your reference list. If you have deposited data etc in the Dryad repository, please include both the 'For publication' link and 'For review' link at this stage.
- If you are requesting an article processing charge waiver, you must select the relevant waiver option (if requesting a discretionary waiver, the form should have been uploaded at Step 3 'File upload' above).
- If you have uploaded ESM files, please ensure you follow the guidance at <https://royalsociety.org/journals/authors/author-guidelines/#supplementary-material> to include a suitable title and informative caption. An example of appropriate titling and captioning may be found at https://figshare.com/articles/Table_S2_from_Is_there_a_trade-off_between_peak_performance_and_performance_breadth_across_temperatures_for_aerobic_scope_in_teleost_fishes_/3843624.

Author's Response to Decision Letter for (RSOS-210392.R0)

See Appendix B.

RSOS-210392.R1 (Revision)

Review form: Reviewer 1

Is the manuscript scientifically sound in its present form?

Yes

Are the interpretations and conclusions justified by the results?

Yes

Is the language acceptable?

Yes

Do you have any ethical concerns with this paper?

No

Have you any concerns about statistical analyses in this paper?

No

Recommendation?

Accept as is

Comments to the Author(s)

The authors have addressed my concerns, and I do not have any further comments.

Review form: Reviewer 2**Is the manuscript scientifically sound in its present form?**

Yes

Are the interpretations and conclusions justified by the results?

Yes

Is the language acceptable?

Yes

Do you have any ethical concerns with this paper?

No

Have you any concerns about statistical analyses in this paper?

No

Recommendation?

Accept as is

Comments to the Author(s)

The authors have addressed all my previous comments.

Decision letter (RSOS-210392.R1)

Dear Dr Xu,

It is a pleasure to accept your manuscript entitled "Predicting the Risk of Rupture for Vertebral Aneurysm based on Geometric Features of Blood Vessels" in its current form for publication in Royal Society Open Science. The comments of the reviewer(s) who reviewed your manuscript are included at the foot of this letter.

Please ensure that you send to the editorial office an editable version of your accepted manuscript, and individual files for each figure and table included in your manuscript. You can send these in a zip folder if more convenient. Failure to provide these files may delay the processing of your proof.

You can expect to receive a proof of your article in the near future. Please contact the editorial office (opencscience@royalsociety.org) and the production office (opencscience_proofs@royalsociety.org) to let us know if you are likely to be away from e-mail

contact – if you are going to be away, please nominate a co-author (if available) to manage the proofing process, and ensure they are copied into your email to the journal.

on behalf of Dr Peter Stewart (Associate Editor) and Mark Chaplain (Subject Editor)
openscience@royalsociety.org

Associate Editor Comments to Author (Dr Peter Stewart):

Thank you for submitting this revision and responding to the comments of the referees. Both referees are now satisfied with the revisions and responses, and so I am pleased to be able to recommend acceptance of your paper in its current form.

Reviewer comments to Author:

Reviewer: 1

Comments to the Author(s)

The authors have addressed my concerns, and I do not have any further comments.

Reviewer: 2

Comments to the Author(s)

The authors have addressed all my previous comments.

Appendix A

We would like to thank all referees for their very helpful comments and suggestions. We extensively revised our paper to address all of them in detail. In what follows, we provide our responses to all individual comments and describe in detail specific changes in the revised manuscript made in response to the specific referees' comments and suggestions.

Responses

Reviewer comments to Author:

Reviewer: #1

Comments to the Author(s)

In this study, Li et al studied the geometric features of vertebral aneurysm for predicting the risk of its rupture. They first extracted 382 non-aneurysm-related features at the distal and proximal parts of the blood vessel, and then the feature selection, and finally the classification using 4 different well-developed ML models. This study tried to address a challenging and yet unsolved research question on predicting rupture risk through geometrical features. While because of a lack of samples, only 37 subjects are involved in this study, the applicability of this would require a large cohort of patient data. This study is closely related to the previous study by Zhao et al. My major concerns are

Comment: *In their previous study, Zhao et al, studied 5 geometric features of VAFA morphology and achieved state-of-the-art classification performance. It is necessary to compare the geometric features proposed in this study, what are the differences, and what about combining the two sets of features together? My suggestion is to conduct a thorough study by including both the geometric features from their previous study and the features beyond aneurysm proposed in this study, then it would address the geometrical features of VAFA morphology properly.*

Response: This study is an attempt to shift from using aneurysm-specific features for segmentation to using more general and automatic data such as the cross-sectional area of the blood vessel. This allows the method to be better applicable when dealing with cases where properties of aneurysm itself change over time or over events such as rupture. Regarding geometric features, aneurysm-specific features from the previous study are not included for the same reason. Because of limited number of samples, we try to minimize the number of features to avoid over fitting. Therefore, we did not use the non-aneurysm specific features from our previous study as they are no drastically different from the ones use in this work.

Comment : *It is necessary to share the codes for reproducibility. I would also suggest sharing the 3D geometrical data if they can so that it will benefit the whole community.*

Response: We will share the code and derived features through GitHub link. Unfortunately, we cannot share the original image data due the hospital data protection policy.

The following Data Access Statement is added

“Data Access Statement

Data and relevant code for this research work are stored in GitHub: [https://github.com/RPTS/Aneurysm 2020 Code](https://github.com/RPTS/Aneurysm_2020_Code), and have been archived within the Zenodo repository: <http://doi.org/10.5281/zenodo.4586810>.”

Comment: *Line 14, they mentioned 'a new segmentation method', the authors can make it clear it is not about image segmentation.*

Response: The original statement has been revised as ‘Based on cross sectional area change of the vessel, we proposed a new main blood vessel segmentation method to generate features from data points.’ to avoid ambiguity in section 1, page 2 of the updated version.

Comment: *Section 2. about the segmentation, it is not clear to me how they use cross-sectional area first-order difference, curvature, and torsion as potential criteria for re-segmentation. Detailed procedures and quantitative criteria shall be provided. In the followed paragraph, the authors have tried to explain but seems very human-dependent. The authors can also perform some uncertainty quantification on the thresholds and related parameters, and also inter-observer and intro-observer variability. The reliability and reproducibility of the segmentation would be very essential for the geometrical feature extraction.*

Response: Sorry for confusion here. In the current work, we only used the cross section area variation for the segmentation. In section 2.3, page 3 of the updated version , it is explained as follows: “We set each segment to have an equal number of curve-fitting points. The thresholds that determine the length of segmentation are set by the standard deviation of cross sectional area. Standard deviation is used here as a statistically meaningful point that determines whether the difference between points and the mean is noteworthy. Our approach is similar to the method for detecting change point for time series data.”

Comment: *Section 2.4, please explain how the integration is done mathematically along the curve. It is not clear 'The ratio between values, along with the angle and distance between two centerline tangent vectors at the neck of the bulge', considering providing a schematic illustration*

Response:

The definition of integral has been included in section A.4.4, page 24. A short explanation of integral has also been provided in the main text in section 3.3.2 page 10.

3.3.2 Curvature

The integral of curvature over segment Proximall (Curvature_integration_P is another important feature for our classification method. The integral is calculated using the method of trapezoidal numerical integration in Matlab, and each segment has a corresponding integral of curvature over arc length (details of formula see Appendix A.4.4). Similar to the investigation of Eccentricity_std_P2, (ruptured, un-ruptured) pairs with significant difference in Curvature_integration_P are chosen, and curvature is plotted against arc length of

The integral is calculated using the method of trapezoidal numerical integration in Matlab, and each segment has a corresponding integral of curvature over arc length. The formula used by the Matlab trapz method is as follows:

(1) For an integration with evenly distributed points, the formula is

$$\int_a^b f(x)dx \approx \frac{b-a}{2N} \sum_{n=1}^N (f(x_n) + f(x_{n+1}))$$

$$= \frac{b-a}{2N} [f(x_1) + 2f(x_2) + \dots + 2f(x_N) + f(x_{N+1})],$$

(2) If the spacing between the N+1 points is not constant, then the formula generalizes to:

$$\int_a^b f(x)dx \approx \frac{1}{2} \sum_{n=1}^N (x_{n+1} - x_n) [f(x_n) + f(x_{n+1})],$$

Based on the segmentation, we generated three groups of features:

- (1) the maximum, mean, standard deviation, integration and variation of the geometric index in each segment;
- (2) the ratios of features in (1) between different segments;
- (3) the angle (α) and distance (d) between two centerline tangent vectors at the neck of the bulge (see Figure 2(b)).

These updates are included in section 2.4, page 3 of the updated version. Illustration is as follows, and is also included in the original paper as Figure 2(b).

Comment: section 2.6, more details can be added and discussed. For example, in SVM, the reason of choosing a linear kernel, what is the performance using a nonlinear kernel, similar for the logistic regression. In the K-Nearest Neighbours, studying different k will be helpful for choosing the k value. What is the majority vote? The choice of random tree depth needs more analysis.

Response: We have conducted experiments using, for example, a different kernel and a different k value. For example, if we were to use an rbf kernel for SVM or k=3 for KNN, then the leave-one-out accuracy would both be 83.8%. We presented the results for choosing a linear kernel and k=5 in the paper since they were shown to yield better results. The explanations for majority vote and the choice of tree depth have been added to section 2.6 as follows:

“It is always possible to keep splitting the data until all points are correctly classified. However, this leads to overfitting. One effective way to prevent this is to control the maximum tree depth. As mentioned, we limited our feature set size to 5. We aimed to minimize the possibility of overfitting while granting a sufficient amount of freedom to the selection process. A maximum tree depth of 1 would be too restrictive since it forces the algorithm to select at most 1 feature and do 1 split. Similarly, a maximum tree depth of 2 implies a total of at most 3 features / splits. Therefore, we set a limit on the tree depth such that using each feature once would be still feasible while overfitting could be avoided as much as possible. For feature sets of size 4 or 5, we set the maximum tree depth to be 3; for smaller feature sets, we set the maximum tree depth to be 2. Hence, we used `sklearn.tree.DecisionTreeClassifier` with its max depth adjusted to the corresponding value.”

Comment: *please add the implementation details of the feature selection and modeling*

Response: Details have been included in section 2.5, page 4 of the updated version. Example: “We used the scikit-learn Python package for implementation, specifically `SelectKBest` and `RFE` from `sklearn.feature_selection`. We specified the estimator as described above and the number of features to select as 5, and we accepted the default value of all the other parameters”.

Comment: *In the results, the authors mentioned the 3 segments and 7 segments. An illustration of how those segments are determined will be helpful.*

Response: An illustration has been included as Appendix Figure A.5, in section A.4, page 19 of the updated version.

Comment: *For Table 1, I would suggest the authors stack the feature ranking together using a bar chart, so that it is much easier to spot the importance of each feature*

Response: Thanks for the suggestion. The following chart figure is much easier to spot the importance of each feature. However, by using a table and bold fonts, the key features are pronounced. At same time, by using the underline, the selected features are also indicated.

Comment: page 7, line 6, what is the unit for the diameter?

Response: The unit is “millimeter”. In the paper, we revised it as follows

“ • a smaller diameter (unit mm) at distal 2 endpoint; ”

Comment: Discussions are more like results. Please merge with the result section, only discussing their implications in the discussion section, including uncertainties, etc.

Response: Thanks for the suggestion. The original discussion has been merged into the results section as section 3.3. Since the implications are tightly linked to each feature listed, remarks are also added to the original discussion session to note related concerns.

Comment: Figure 5, please add units for the horizontal and vertical axis. The total cases are 37, then what is case 42? What is the meaning of the error bar in (c), is the mean eccentricity constant? Figure (c) is small and hard to see. Figures 6, 7, and 8 can be placed into the appendix.

Response: The axes represent the x,y,z coordinates of the curve-fitting points and the units are mm. Edits have been added to the figures. “Case 42” is the raw data index. In the updated version, cases have been reindexed.

(a)

(b)

For eccentricity specifically for figure 5, we calculated the eccentricity for the second proximal segment of case 27, and that of case 18, respectively. The error bars place each point at the center of the vertical bar, with lengths of each error bar above and below the data points determined by how far each eccentricity point deviates from the mean eccentricity. The mean eccentricity is constant for a specific segment of a specific case, but differs across segments and cases. These clarifications have been added to the paper.

Figure 7 and 8 are placed into the appendix, whereas Figure 5 and Figure 6 are preserved in the main text for the purpose of comparison. Figure (c) is enlarged for every figure.

Comment: Figure 9 has a similar issue as Figure 5. Please define the integral of curvature mathematically, it is not clear at (c) how the integral varies along the arc length. Figures 10 and 11 can be placed in the Appendix.

Response: The integral is calculated using the method of trapezoidal numerical integration in Matlab, and each segment has a corresponding integral of curvature over arc length. This clarification has been added to the paper in section 3.3.2. The integral does not vary along the arc length, but it partially captures how curvature varies as mentioned in the paper. Figures 10 and 11 have been placed in the Appendix.

Comment: Figure 12, similar issue. Figure 13 can be included in the appendix

Response: Figure 13 is placed into the appendix.

Comment: It would be helpful if the authors can include the definitions of some geometric indices into the main text, rather than in the appendix, to make the paper self-contained. For example, the

curvature and its integration, etc. The Figure in the appendix should be renumbered, not following the main text. A.2 should be included in the main text for the convenience of reading. Figure 19(b) is not exactly the convex hull of the blue curve.

Response: Definitions of frequently-mentioned geometric features such as curvature, torsion, and eccentricity are included in the main text in section 2.2, page 2 (with Figure A.2.) , while the full list of definitions remains in the appendix. The figures in the appendix are renumbered. Figure 19(b) has been revised.

Reviewer: #2

Comments to the Author(s)

Overview

The present manuscript is devoted to the investigation of various morphological features to predict rupture. The same 37 fusiform cases are studied as already published by the present authors in 2018 (Zhao et al., 2018 Vertebral artery fusiform aneurysm geometry in predicting rupture risk. Royal Society Open Science 5(10), 180780. (doi:10.1098/rsos.180780)). The present manuscript is incremental compared to this previous work by the authors. Therefore, I do not recommend the publication of the present manuscript in Royal Society Open Science.

Response: There is a significant difference between this paper and our previous study, where aneurysm related features are used for classification. In the present study, we have removed aneurysm related features and attempted to answer the following question: is it sufficient to predict rupture by using only geometric information of the vessels? From the medical point of view, this will give doctors a much more useful predictive tool since they normally only focus on aneurysm features.

Major comments:

Comment: *Further details regarding the segmentation should be provided, including methods and smoothing. Some of the investigated morphological features are highly sensitive on these segmentation steps.*

Response: We set each segment to have an equal number of curve-fitting points. The thresholds that determine the length of segmentation are set by the standard deviation of cross sectional area. More details have been added to section 2,3, page 3 of the update version.

Comment: *The authors state that the blood flow essential for the rupture prediction of brain aneurysm. No flow simulations are involved in the present study, only morphological parameters are investigated. It is not clear how the distal parts influence the blood flow (Figures 12 & 13).*

Response: The flow of blood in vessels is essential to the rupture of brain aneurysm. The geometric properties of the proximal vessels could potentially impact the normal blood flow, and those of the distal vessels may reflect such change upstream. Changes have been added to section 2,3, page 3 of the update version

Comment: *The number of the investigated cases is relatively low to obtain representative results.*

Response: This study is an attempt to shift from using aneurysm-specific data for segmentation to using more general and automatic data such as the cross-sectional area of the blood vessel. It is also a starting point where applicable methods are developed to deal with different cases. Following this study, more related data can be obtained and used to enhance the model in the future.

Minor comments:

Comment: *British (e.g. modelling, centre) and US spelling (e.g. categorized, organized, modeling, optimized, center) are mixed up. Only one of the spelling should be used in the manuscript.*

Page 4, line 42: the -> The

Page 13, line 26: Aneurysms -> aneurysms

Response: Thanks for careful reading. We have revised the typos.

Appendix B

We would like to thank all referees for their very helpful comments and suggestions. We extensively revised our paper to address all of them in detail. In what follows, we provide our responses to all individual comments and describe in detail specific changes in the revised manuscript made in response to the specific referees' comments and suggestions.

Responses

Reviewer comments to Author:

Reviewer: #1

Comment 1: *The authors have tried to address all my comments, thanks for their efforts. While some of the comments are not fully resolved, thus I do not recommend for publication of the current revision. They are*

(1) Original comment 1: I am not convinced that the aneurysm-specific features can be ignored. It is necessary to have a comparison among these three combinations: (1) only aneurysm-specific features, (2) only use the non-aneurysm-specific features, and (3) the combinations of the above two features. This will be interesting to the readers and also make the study completed, rather than incremental and somehow disconnected from their previous study.

Response: Thanks for the great suggestion. In our previous study, we followed the clinic doctors' comment intuition and thought aneurysm-specific features should play important roles on the rupture risk. However, when we used the aneurysm-specific features, we discovered that 4 out of the 5 top features involve characteristics of surrounding blood vessels. Consequently, we hypothesized that parent blood vessels may play more important roles on rupture and can be used for rupture prediction independently. This motivated the new segmentation and feature processing methods in this study.

We in fact found that aneurysm-specific features have little significance in comparison to the features in our new dataset. Specifically, only two aneurysm-specific features -- Solidity_integration and Eccentricity_integration -- pass the t-test with a 0.1 threshold. Using the combined dataset, the top 2 features from each of the four feature selection algorithms described in the paper are as follows:

- SelectKBest: Curvature_mean_PD, CrossArea_mean_PD2
- RFE(SVM): Curvature_mean_PD, Curvature_integration_P
- RFE(LR): Curvature_mean_PD, Eccentricity_std_P2
- RFE(RF): Diameter_normalD2, Curvature_mean_PD

This list of top features is exactly the same as the list obtained using the non-aneurysm-specific dataset. The feature selection and model training results are not affected, showing the sufficiency of the currently selected blood vessel features.[1] More detailed result is attached at the end of the response.

This result confirms our hypothesis that geometry of parent blood vessels is also quite an important factor for fusiform aneurysms rupture. This is also consistent with previous studies on hemodynamics of aneurysms that the parent vessel of the aneurysm is strongly associated with hemodynamics characteristics which in turn contributes to

aneurysm formation, growth and rupture.

```

▶ # combined dataset including aneu and non-aneu features
fs_table = feature_selection(path='AllFeatures1.csv')

33 features pass t-test (pval<=0.1):
Index(['Eccentricity_integration_A', 'Curvature_mean_PD',
      'Curvature_integration_P', 'LineDist', 'Diameter_normalD2',
      'Torsion_absmean_D2', 'Curvature_mean_P', 'Solidity_integration_PD2',
      'Torsion_variation_D2', 'Torsion_mean_P2', 'Ratio_Dpnordnor2',
      'Solidity_integration_A', 'MinorAxis_std_P2', 'MajorAxis_mean_PD2',
      'Eccentricity_integration_PD2', 'Curvature_absmean_D2',
      'Curvature_mean_D2', 'EquipDiameter_mean_PD2', 'Eccentricity_std_P2',
      'EquipDiameter_max_PD2', 'PointDist', 'CrossArea_mean_PD2',
      'MajorAxis_mean_D2', 'EquipDiameter_mean_D2', 'MajorAxis_max_PD2',
      'Curvature_variation_D2', 'MinorAxis_mean_PD2', 'Torsion_std_D2',
      'MinorAxis_max_PD2', 'CrossArea_max_PD2', 'Eccentricity_variation_D2',
      'MinorAxis_mean_D2', 'Ratio_Dpnordnor1'],
      dtype='object')
results from feature selection:
      SelectKBest
      Xs freqs
0      Curvature_mean_PD      26
1      CrossArea_mean_PD2      17
2      Curvature_integration_P      13
3      CrossArea_max_PD2      9
4      Ratio_Dpnordnor1      8
5      Curvature_mean_P      8
6      MajorAxis_mean_PD2      7
      RFE(SVM)
      Xs freqs
0      Curvature_mean_PD      23
1      Curvature_integration_P      22
2      Eccentricity_std_P2      19
3      LineDist      17
4      Diameter_normalD2      15
5      Ratio_Dpnordnor1      8
6      PointDist      6
      RFE(LR)
      Xs freqs
0      Curvature_mean_PD      28
1      Eccentricity_std_P2      22
2      Curvature_integration_P      21
3      Ratio_Dpnordnor1      17
4      LineDist      16
5      Diameter_normalD2      9
6      Eccentricity_variation_D2      6
      RFE(RF)
      Xs freqs
0      Diameter_normalD2      24
1      Curvature_mean_PD      20
2      Eccentricity_std_P2      18
3      Ratio_Dpnordnor1      14
4      Eccentricity_variation_D2      10
5      MinorAxis_std_P2      9
6      EquipDiameter_max_PD2      8

```

In the revised version, pages 16-17 discussion session we added the several paragraph as follows

In the previous study [5], authors followed the clinic doctors' common intentions that aneurysm-specific features should play important roles on the rupture risk. However, when we used the aneurysm-specific features, we discovered that 4 out of the 5 top features involve characteristics of surrounding blood vessels. Consequently, we hypothesized that parent blood vessels may play more important roles on rupture and can be used for rupture prediction independently. In fact, if we add the aneurysm-specific features to the features in our new dataset, only two aneurysm-specific features: 'Solidity_integration' and 'Eccentricity_integration' – pass the t-test with a 0.1 threshold. Using the combined dataset, the top 2 features from each of the four feature selection algorithms described in the paper are as follows:

- SelectKBest: Curvature_mean_PD, CrossArea_mean_PD2
- RFE(SVM): Curvature_mean_PD, Curvature_integration_P
- RFE(LR): Curvature_mean_PD, Eccentricity_std_P2
- RFE(RF): Diameter_normalD2, Curvature_mean_PD

This ranking list is exactly the same as the ranking list using the non-aneurysm-specific dataset. This result confirms our hypothesis that geometry of parent blood vessels is also quite an important factor for fusiform aneurysms rupture. This is also consistent with previous studies on hemodynamics of aneurysms that the parent vessel of the aneurysm is strongly associated with hemodynamics characteristics which in turn contributes to aneurysm formation, growth and rupture [22].

Comment 2: Original Comment 4 about uncertainty quantification: Again this is important for the classification. If the uncertainty from the feature extraction is significantly large, then the classification based on those features could be problematic. The authors seem to have not addressed or discussed this concern in the revision.

Response:

Thanks for the feedback. A detailed pseudo-code of the algorithm is included in the paper under Section 2.3 in Page 3. More explanations are as follows:

The segmentation process utilizes points on the centerline curve. We set each segment to have an equal number of curve-fitting points. The thresholds that determine the length of segmentation are set by the standard deviation of cross sectional area. Standard deviation is used here as a statistically meaningful point that determines whether the difference between points and the mean is noteworthy. Our approach is similar to the method for detecting change points for time series data. The detailed algorithm is summarized as follows (also see Figure2 (c)). Define:

- a = label of aneurysm starting point
- b = label of aneurysm end point
- s = the number of curve fitting points within a segment
- n = the total number of curve fitting points for this individual
- ps = cross sectional area at start of aneurysm
- pe = cross sectional area at the end of aneurysm

To calculate s, we further define:

- baseline = ps (if the point of interest is in the proximal vessel) and pe (if the point of interest is in the distal vessel)
- threshold = standard deviation of cross sectional area of the entire vessel (excluding the aneurysm part)
- $default_s = \min((a - 1)/2, (n - b)/2)$ ¹

¹the maximum number of points a segment can have

Algorithm 1: Calculation of s

```

1: INPUT: Centerline datapoints:  $\mathcal{D}$ , and all parameters above;
2: scan the proximal vessel to find the first point whose difference with baseline cross sectional area exceeds the threshold :  $p1$  ;
   for point  $i$  in  $D[a:1:1]$  do
   |   find the first point  $p1$  in the proximal vessel that satisfies  $|p1 - baseline| \geq threshold$  ;
   end
3: scan the distal vessel to find the first point whose difference with baseline cross sectional area exceeds the threshold :  $d1$  ;
   for point  $i$  in  $D[b:1:n]$  do
   |   find the first point  $d1$  in the distal vessel that satisfies  $|d1 - baseline| \geq threshold$  ;
   end
4: calculate s based on  $p1$  and  $d1$  ;
   if  $p1$  and  $d1$  both can be found then
   |    $s = \min(a - p1, d1 - b)$  ;
   end
   if only  $p1$  can be found and  $(a - p1) \leq default_s$  then
   |    $s = a - p1$  ;
   end
   if only  $d1$  can be found and  $(d1 - b) \leq default_s$  then
   |    $s = d1 - b$  ;
   end
   if neither  $p1$  nor  $p2$  can be found then
   |    $s = default_s$  ;
   end
5: OUTPUT:
   the number of points for each segment = s, the label of the first proximal point = a-s,
   the label of the second proximal point = a-2s,
   the label of the first distal point = b+s,
   the label of the second distal point = b+2s ;

```

The detection of points exceeding the threshold is entirely conducted by the algorithm. We recognize that there might be uncertainty brought by different definitions of a and b, which are the start and end point of the aneurysm. By using exactly the same data determined by doctors in the previous paper, we hope to preserve the consistency between this paper and the previous paper, which allows us to better compare the two results.

In the stated algorithm, we used the standard deviation of the cross sectional area at non-aneurysm points as a threshold. The threshold served as the basis for evaluating the variation of cross sectional area at these points compared to the baseline, and choosing segmentation points. To examine how the choice of threshold influences the final result, we applied different multipliers - 0.9 and 1.1 - to the current threshold. The selected features based on different threshold values are illustrated below.

As described in the paper, we considered two methods for selecting top features: one is to take the union set of the top 2 features of each selection algorithm, the other is to take the set of features with a frequency of at least 20 (out of 30) times. The following table compares the top features selected under different segmentation thresholds:

multiplier	1.0		0.9		1.1	
	top2	≥20	top2	≥20	top2	≥20
Curvature_mean_PD	x	x	x	x	x	x
Diameter_normalD2	x	x	x	x	x	x
Eccentricity_std_P2	x	x			x	x
Curvature_integration_P	x	x	x	x		
CrossArea_mean_PD2	x		x	x	x	
Solidity_integration_PD2					x	x
MajorAxis_mean_D2			x	x		

There is a great amount of overlap between the sets using the different thresholds, supporting stability of the algorithm.

In addition, the following table compares the best accuracy achievable using a set of two features, specifically Curvature_mean_PD and Diameter_normalD2. The method we are using is robust since changing the parameter (threshold) has small effects as shown by our numerical experiments.

	1.0		0.9		1.1	
	LOO	80/20	LOO	80/20	LOO	80/20
SVM linear	83.8	81.8	75.7	77.1	86.5	84.5
Logistic Regression	83.8	81.8	81.1	80.5	89.2	85.6

Reviewer: #2

Comment 1: *The authors have virtually removed the aneurysms in the investigated cases and they try to find characteristic features of the vessel geometries in order to predict the risk of the aneurysm rupture. This is certainly interesting; however the blood vessel and the features of the aneurysms should be treated simultaneously.*

Response: Thanks for the suggestion. As the answer in the comment 1 of the first reviewer.

In the revised version, pages 16-17 discussion session we added the several paragraph as follows

In the previous study [5], authors followed the clinic doctors' common intentions that aneurysm-specific features should play important roles on the rupture risk. However, when we used the aneurysm-specific features, we discovered that 4 out of the 5 top features involve characteristics of surrounding blood vessels. Consequently, we hypothesized that parent blood vessels may play more important roles on rupture and can be used for rupture prediction independently. In fact, if we add the aneurysm-specific features to the features in our new dataset, only two aneurysm-specific features: 'Solidity_integration' and 'Eccentricity_integration' – pass the t-test with a 0.1 threshold. Using the combined dataset, the top 2 features from each of the four feature selection algorithms described in the paper are as follows:

- SelectKBest: Curvature_mean_PD, CrossArea_mean_PD2
- RFE(SVM): Curvature_mean_PD, Curvature_integration_P
- RFE(LR): Curvature_mean_PD, Eccentricity_std_P2
- RFE(RF): Diameter_normalD2, Curvature_mean_PD

This ranking list is exactly the same as the ranking list using the non-aneurysm-specific dataset. This result confirms our hypothesis that geometry of parent blood vessels is also quite an important factor for fusiform aneurysms rupture. This is also consistent with previous studies on hemodynamics of aneurysms that the parent vessel of the aneurysm is strongly associated with hemodynamics characteristics which in turn contributes to aneurysm formation, growth and rupture [22].

Comment 2: *Have the authors compared the cases after removing the aneurysms with healthy subjects? It would be interesting to see the blood vessel structures for aneurysm patients as well as for healthy subjects. The presented tools might predict the risk of developing an aneurysm for certain blood vessels.*

Response: This is a really great suggestion. It is very meaningful to compare with the healthy vessel structures. This is in fact what we hope to do as a next step. We are

prospectively collecting the contralateral normal artery data for an autologous control study. In the revised version, we added a paragraph in the discussion section

“At the same time, our results confirmed that the parent vessel plays an important role in fusiform aneurysm rupture. We are prospectively collecting the contralateral normal artery data for an autologous control study. In the future, we will generalize current work and develop a model to predict aneurysm formation risk by comparing the blood vessel structures for aneurysm patients as well as for healthy subjects.”

Comment 3: *The following statement should be refined "The geometric properties of the proximal vessels could potentially impact the normal blood flow, and those of the distal vessels may reflect such change upstream." (Section 2.3) The question is not whether the flow is normal or not. It is more important to realize that the blood flow is highly influenced by the vessel geometry, e.g., the inflow-jet in the aneurysm is absolutely determined by the vessel. Unfortunately, these hemodynamics effects are missing from the present study.*

Response: Thanks for pointing it out. In the revised version, it is changed to be “The hemodynamic [Cerebral] plays an essential role in the rupture of brain aneurysm. The geometric properties of the proximal vessels impact the normal blood flow, and those of the distal vessels also reflect such change upstream [Sforza].”

Thanks for the suggestion. Yes, we do not include the hemodynamics explicitly in this work. This is the limitation of morphological study. The current study attempts to shed a different light on the topic by exploring the geometric properties of the vessels, and propose an efficient & accurate auxiliary diagnosis and treatment method. Doctors could make quick decisions by the vessel images without doing full CFD. Also, identifying the most important geometry parameters could help us to understand the mechanism deeper by hemodynamics simulation.